# A broadly neutralizing anti-influenza antibody reveals ongoing capacity of haemagglutinin-specific memory B cells to evolve

Ying Fu[1], Zhen Zhang[1], Jared Sheehan[1], Yuval Avnir[1], Callie Ridenour[2], Thomas Sachnik[1], Jiusong Sun[1], M. Jaber Hossain[2], Li-Mei Chen[2], Quan Zhu[1], Ruben O. Donis[2] & Wayne A. Marasco[1]

Understanding the natural evolution and structural changes involved in broadly neutralizing antibody (bnAb) development holds great promise for improving the design of prophylactic influenza vaccines. Here we report an haemagglutinin (HA) stem-directed bnAb, 3I14, isolated from human memory B cells, that utilizes a heavy chain encoded by the *IGHV3-30* germline gene. MAb 3I14 binds and neutralizes groups 1 and 2 influenza A viruses and protects mice from lethal challenge. Analysis of VH and VL germline back-mutants reveals binding to H3 and H1 but not H5, which supports the critical role of somatic hypermutation in broadening the bnAb response. Moreover, a single VLD94N mutation improves the affinity of 3I14 to H5 by nearly 10-fold. These data provide evidence that memory B cell evolution can expand the HA subtype specificity. Our results further suggest that establishing an optimized memory B cell pool should be an aim of 'universal' influenza vaccine strategies.

[1] Department of Cancer Immunology and Virology, Dana-Farber Cancer Institute, Harvard Medical School, 450 Brookline Avenue, Boston, Massachussetts 02215, USA. [2] Influenza Division, Centers for Disease Control and Prevention, National Center for Immunization and Respiratory Diseases, 1600 Clifton Road-Mail Stop G-16, Atlanta, Georgia 30333, USA. Correspondence and requests for materials should be addressed to W.A.M. (email: wayne_marasco@dfci.harvard.edu).

nfluenza viruses are a main cause of acute respiratory illness in human and many animal species. Seasonal influenza viruses infect 5–15% of the population worldwide annually, which results in 250,000–500,000 deaths[1]. Pandemic influenza strains cause less frequent but severe global outbreaks and can be responsible for significant morbidity and high mortality, especially among healthy, young adults. The most infamous example of pandemic influenza, the 'Spanish Flu,' killed at least 40 million people in 1918–1919 (refs 2,3).

Influenza viruses are characterized by segmented negative sense RNA genomes. On the basis of their antigenic differences in the virion core proteins, they are divided into three main types: A, B and C. Influenza A viruses are the most pathogenic in humans and are further subclassified by the two major surface proteins: haemagglutinin (HA) and neuraminidase (NA). HA is responsible for binding to host sialic acid glycan receptors, mediating cell entry and viral RNA release to the cytoplasm, whereas NA is critical for nascent virion budding out of host cells by cleaving sialic acid[4,5]. There are 18 HA subtypes and 11 NA subtypes, which make up all known influenza A viruses by various combinations of HA and NA[1–3,6]. Furthermore, based on the phylogenetic relationships of HA genes, the 18 HA serotypes are classified into two major groups: groups 1 and 2 (ref. 7) (Fig. 1a). As in all RNA viruses, the low-fidelity of influenza virus polymerases result in high mutation rates[8,9]. Mutations in the HA and NA genes often impart antigenic changes, known as antigenic drift, that mediate evasion of host immune response by seasonal viruses[10]. In addition, genetic re-assortment between seasonal and animal influenza viral genomes can yield viruses with novel antigenic characteristics that would not be susceptible to human population immunity elicited by seasonal viruses. Human-to-human transmission of such viruses lead to occasional worldwide pandemics[8]. The recent outbreak of avian H7N9 influenza virus in China has resulted in a total of 681 laboratory-confirmed cases and at least 275 deaths reported to WHO, posing a rapidly growing pandemic threat to public health[11].

The rapid antigenic evolution of seasonal influenza viruses poses a formidable challenge to the development of long-lasting and effective prophylactic or therapeutic strategies against influenza viruses. Currently, influenza vaccination is the most effective disease control intervention but constant surveillance of the antigenic properties of circulating viruses is necessary to update vaccine composition when antigenic drift is detected. Unfortunately, antigenic variants may sporadically emerge after the start of vaccine manufacturing resulting in a poor vaccine match to viruses circulating in the upcoming influenza season. The discovery of human broadly neutralizing antibodies (bnAbs) that target highly conserved epitopes on the stem region of influenza HA has shed light on a potential pathway to 'universal' flu vaccines. In fact, the stem region has become a main target for development of novel treatments using either antibody-based vaccine design or passive immunotherapy. So far, several bnAbs capable of neutralizing multiple serotypes of influenza A virus within group 1 and/or group 2 have been isolated from immunized mice, phage libraries, memory B or plasma cells of immune donors[12–21]. Most bnAbs, such as F10 (ref. 12), CR6261 (ref. 13), MAb 3.1 (ref. 18) and CR8020 (ref. 14), neutralize either group 1 or group 2 influenza viruses. Numerous bnAbs, namely FI6 (ref. 15), CR9114 (ref. 16), 39.29 (ref. 17), MAb 1.12 (ref. 19), CT149 (ref. 20) and 2B06 (ref. 21) are reported to be capable of neutralizing human influenza A viruses from both groups. Moreover, mAb CR9114 can also bind HAs from influenza B lineage and protect against lethal challenge with influenza B viruses in vivo[16]. MAb 1.12 and CR9114 were isolated from human-derived phage display libraries that were each derived from a single seasonal influenza experienced donor[16,19].

Important for the present study, the natural pairs of the variable regions of heavy chain (VH) and light chain (VL) are uncoupled and randomly combined using antibody phage display, which limits the analysis of antibody evolution. MAb FI6 was discovered from approximately 104,000 in vitro cultured human plasma cells from healthy donors shortly after natural infection with influenza A or vaccination[15]. MAb 39.29 was selected from ∼950 IgG+ plasmablasts through antigen-specific sorting and hSCID mice in vivo expansion, which increased the efficiency by ∼100-fold (ref. 17). MAbs CT149 and 2B06 were obtained from HA-specific antibody-secreting cells using a high-throughput cell-based microwell array chip (ISAAC)[20,22] and single-cell PCR with reverse transcription (RT–PCR)[21,23] from individuals who had received seasonal influenza vaccination, respectively.

In humoral immunity, plasma cells are differentiated from naive B cells, long-lived plasma cells and memory B cells[24,25]. In response to viral re-infection and vaccination, long-lived plasma cells produce neutralizing antibodies, specifically recalling the original virus, whereas the memory B cells contribute by producing high-affinity neutralizing antibodies specific for the variant virus by re-entering germinal centers[26,27]. Furthermore, the somatic hypermutations (SHMs) of memory B cells could be accumulated in older individuals through repeated cycles of antibody divergence and selection[28,29]. In this way, memory B cells have a broader repertoire of antigen specificity than long-lived plasma cells[24]. Therefore, it is considered essential for a long-lasting, broadly efficacious vaccine to develop the stable population of memory B cells and elicit potent bnAb responses. In this study we characterize a new bnAb 3I14 isolated from memory B cells that binds to the HA stem domain of both group 1 and group 2 influenza viruses. We performed evolution studies to document the manner by which this 3I14 can broaden its neutralization activity against H5N1 viruses. 3I14 is also the fourth bnAb to be identified that uses the *IGHV3-30* germline gene, which provides additional evidence of a second IGHV germline gene that shows biased use against this critically important stem epitope.

## Results

**Isolation of bnAbs from cultures of single memory B cells.** To isolate bnAbs against influenza viruses from human memory B-cell repertoire, we established a rapid and reliable culture method, which allowed for human memory B-cell activation and differentiation in vitro (Supplementary Fig. 1). Antigen-specific human memory B cells (CD19+CD27+) were isolated from peripheral blood mononuclear cells (PBMCs) of seven healthy donors using tetramerized H3 (A/Brisbane/10/07) trimers; only 0.19–1.08% of memory B cells were reactive with H3 (Table 1). These B cells were sorted into 384-well plates at the density of one cell per well and cultured in the presence of irradiated CD40L-transfected cells. After 14 days, 1051 (39.1% in 2,688 cultures) culture supernatants derived from seven donors were found to secrete IgG or IgM and were sequentially tested for reactivity with H3 (A/Brisbane/10/07), H7 (A/Canada/RV444/04), H1 (A/California/04/09) and HA of influenza B (B/Malaysia/2506/04). Through this screen, 237 (22.55%) expanded memory B cells were found to secrete Igs that bound to H3 (Table 1). This in vitro expansion step resulted in a 37-fold increase in H3-reactive B cell recovery compared with 0.61% recovery by sorting plus RT–PCR only (data not shown). The average percentage of cross-reactive clones within the group 2 strains H3/H7 was 18.14%. Remarkably, 13.08 and 8.44% of the H3-binding clones showed hetero-subtypic binding to group 1 H1 strains and H7/H1 strains, respectively. Only 3.38% H3-reactive (H3+) clones were found to

also bind influenza B. Next, the supernatants of memory B-cell clones that showed hetero-subtypic binding were tested for microneutralization against H3N2 (A/Brisbane/10/07). One bnAb, 3I14, derived from donor 3 that showed H3/H7/H1 cross-reactivity and neutralization was further characterized.

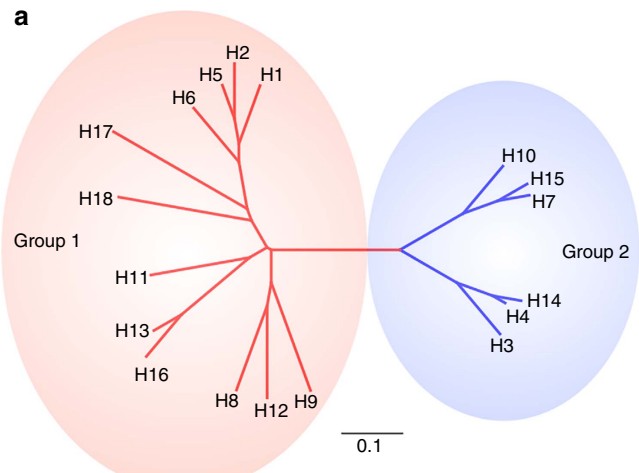

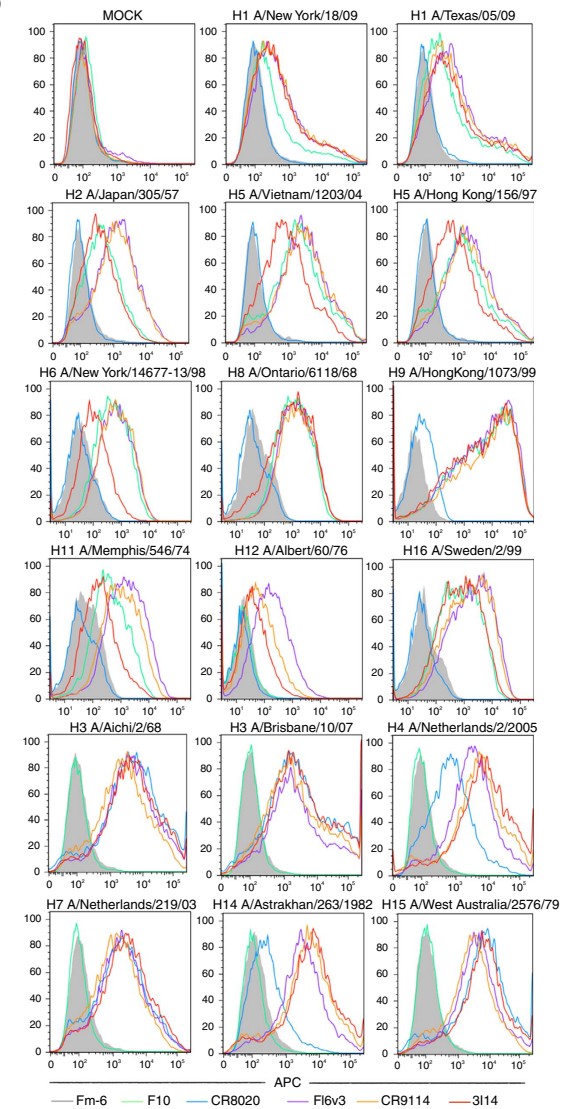

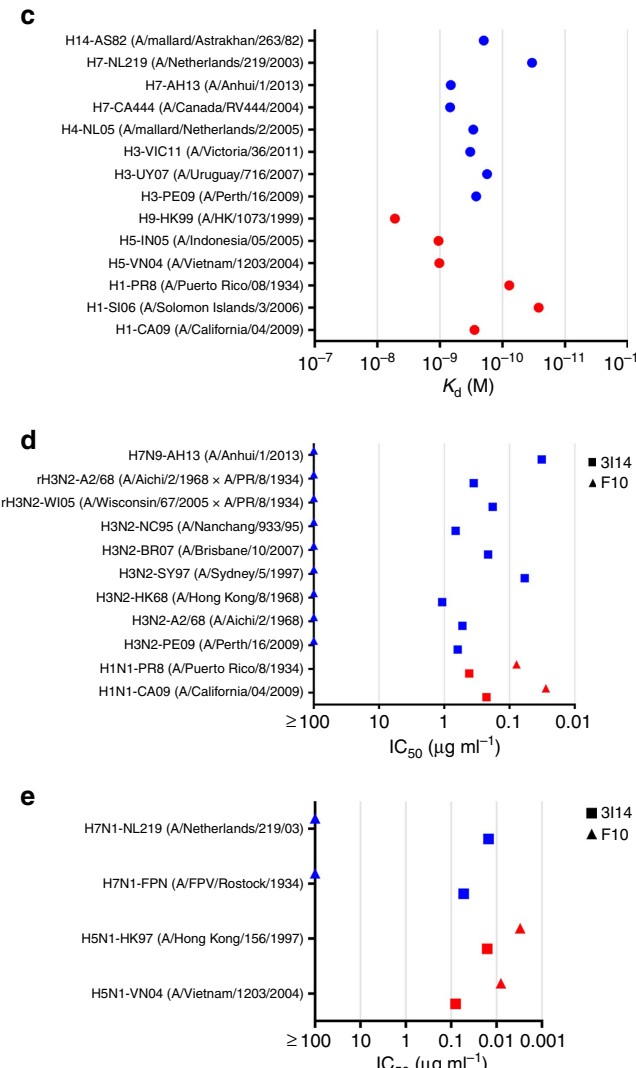

**Figure 1 | Reactivity of 3I14 against group 1 and group 2 influenza A viruses.** (**a**) Phylogenetic tree of the 18 HA subtypes of influenza A viruses based on amino acid sequences. Group 1 subtypes are listed in red and group 2 subtypes in blue. The amino acid distance scale bar denotes a distance of 0.1. (**b**) FACS analysis of 3I14 binding to a broad range of group 1 and group 2 HAs. 293T cells were transiently transfected with different HA-expressing plasmids, followed by staining with the purified scFvFc antibodies and APC-labelled mouse anti-human Fc antibody. Binding of 3I14 (red line), F10 (group 1-specific, green line), CR8020 (group 2-specific, blue line), FI6v3 (groups 1 and 2 specific, purple line), CR9114 (group 1 and 2 specific, orange line), and irrelevant mAb Fm-6 (anti-SARS virus, grey filled histogram) were analysed by flow cytometry. Data represent a representative experiment from three independent experiments. (**c**) 3I14 scFvFc Ab binding ($K_d$ values) to recombinant HAs representative of group 1 (red) or group 2 (blue) subtypes. Data represent a representative experiment from three independent experiments. (**d**) 3I14 scFvFc Ab neutralization ($IC_{50}$ values) of infectious viruses group 1 (red) or group 2 (blue) subtypes. 3I14 was represented by squares; anti-group 1 mAb F10 was represented by triangles. Graphs used for $IC_{50}$ values determined by averaging neutralization titre of two to three independent experiments. (**e**) 3I14 scFvFc neutralization ($IC_{50}$ values) of pseudoviruses representative of group 1 (red) or group 2 (blue) subtypes. These data represent average neutralization titres of two to three independent experiments. Anti-group 1 mAb F10 scFvFc was used for reference. FACS, fluorescence-activated cell sorting.

**Table 1 | Expanded memory B cells (mB) in seven healthy donors.**

| Donor No. | H3$^+$ population of mB (CD19$^+$/CD27$^+$) | Clonable H3$^+$ mB | Clonable H3$^+$ mB cross to H7$^+$ | Clonable H3$^+$ mB cross to H1$^+$ | Clonable H3$^+$ mB cross to H1$^+$/H7$^+$ | Clonable H3$^+$ mB cross to B$^+$ |
|---|---|---|---|---|---|---|
| 1 | 0.94% | 29 | 0 | 1 | 0 | 1 |
| 2 | 0.30% | 32 | 6 | 1 | 0 | 0 |
| 3 | 0.32% | 29 | 8 | 10 | 6 | 0 |
| 4 | 0.19% | 28 | 4 | 2 | 1 | 0 |
| 5 | 0.95% | 31 | 2 | 0 | 0 | 2 |
| 6 | 0.51% | 66 | 20 | 14 | 11 | 5 |
| 7 | 1.08% | 22 | 3 | 3 | 2 | 0 |
| Total | — | 237 | 43 | 31 | 20 | 8 |
| Average percentage (%) | 0.61 | 22.55* | 18.14† | 13.08† | 8.44† | 3.38† |

*Percent clonable mBs are from 1051 Ig positive cultures.
†Percent clonable mBs are from 237 H3 positive cultures.

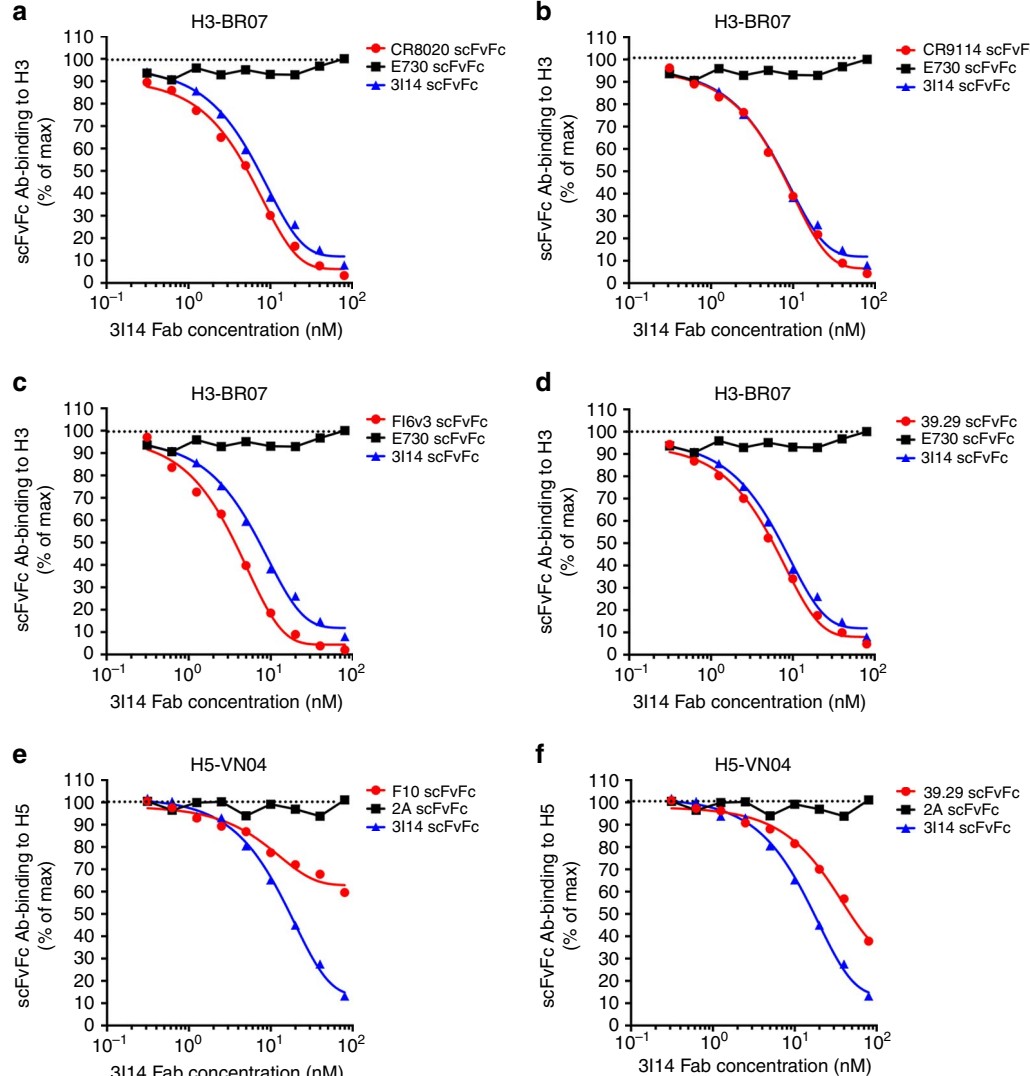

**Figure 2 | 3I14 cross-competes with binding by other anti-stem bnAbs, FI6v3, CR9114, 39.29, F10 and CR8020 to H3 or H5.** 5 µg ml$^{-1}$ H3-BR07 or H5-VN04 protein immobilized on ELISA plates were incubated with twofold serial dilution of 3I14 Fab from 80 to 0.3 nM mixing with other scFvFc Abs at 5 nM. The binding of scFvFc Abs was detected using HRP-conjugated mouse anti-human CH$_2$ antibodies. 3I14 Fab strongly inhibits the binding of CR8020, CR9114, FI6v3 and 39.29 to H3-BR07 but not with E730 (**a–d**). 3I14 partially inhibits the binding of F10 and 39.29 to H5-VN04 but does not inhibit the binding of 2 A (**e,f**). HRP, horseradish peroxidase

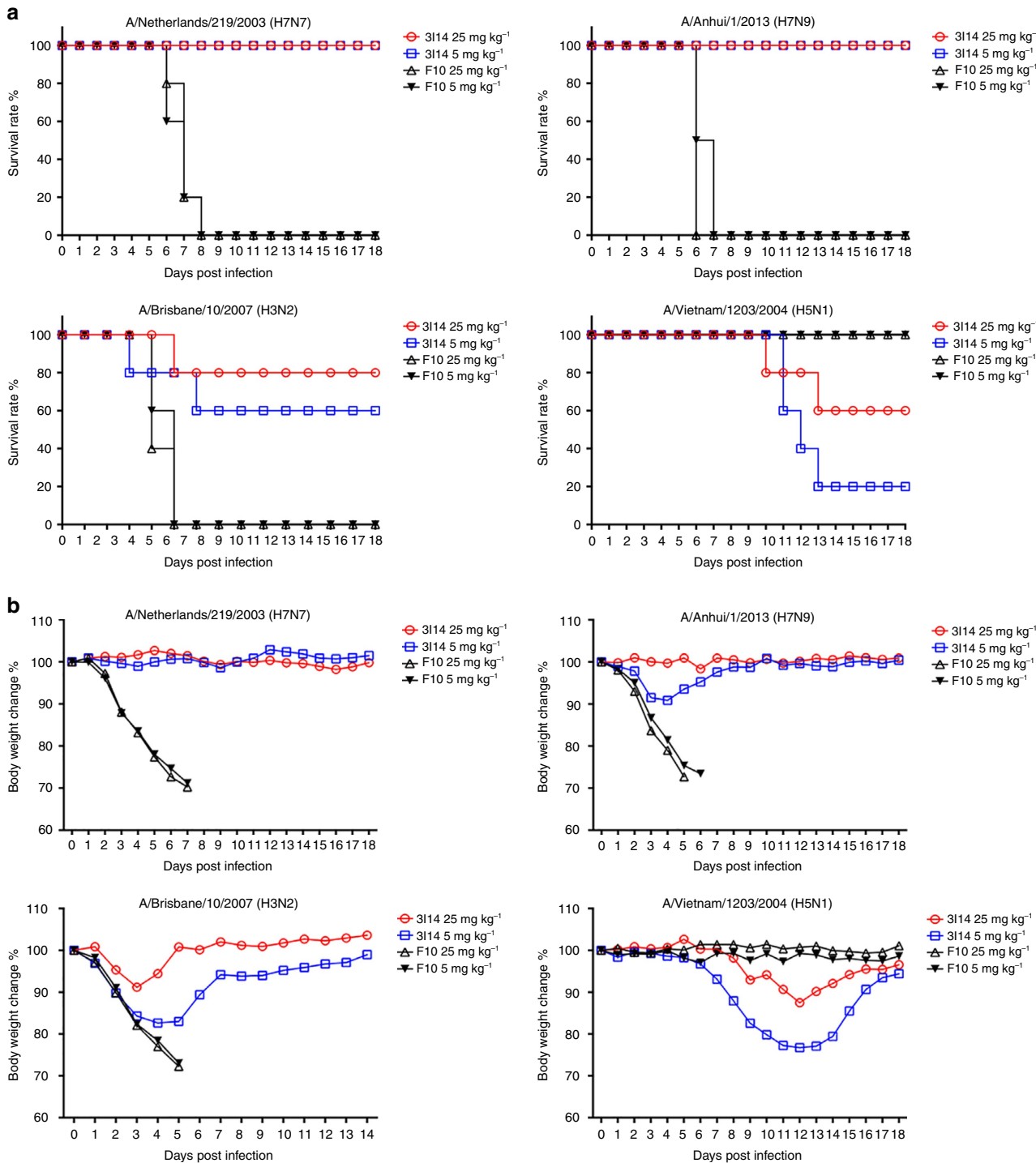

**Figure 3 | Protective efficacy of 3I14 in mice.** Group of five mice were treated with 25 or 5 mg kg$^{-1}$ doses of purified IgGs given intraperitoneally 24 h before lethal challenge by i.n. inoculation with H7N7-NL219, H7N9-AH13, H3N2-BR07-ma or H5N1-VN04 influenza viruses ($\sim$10 LD$_{50}$). (**a**) Survival (%) and (**b**) body weight change (%) of mice that treated with bnAb 3I14 and group 1 control mAb F10. Data represent mean change in body weight of five mice per group compared with their baseline body weight.

**3I14 is a highly mutated *IGHV3-30*-encoded antibody**. The sequences of the VH and VL chains were recovered from the expanded single-cell culture using RT–PCR. 3I14 is encoded by the *IGHV3-30\*18* and *IGLV1-44\*01* germline genes. The rearranged heavy chain possesses a long complementarity determining region 3 (HCDR3) (23 amino acids) and uses the *IGHD3-22\*01* DH segment flanked by large N-additions at both VH and *IGHJ4\*02* junctions (Supplementary Fig. 2). 3I14 mAb has 15 variable heavy chain and seven variable light chain

SHMs excluding the primer-flanking regions, which are observed in both the framework and CDRs.

**3I14 binds and neutralizes groups 1 and 2 influenza viruses**. 3I14 bound cell surface-expressed HAs across serotypes of both group 2 (H3, H4, H7, H14 and H15) and group 1 (H1, H2, H5, H6, H8, H9, H11, H12 and H16) influenza A viruses by flow cytometry (Fig. 1b). 3I14 also bound purified HA proteins of different subtypes that belong to group 2 (H3, H4, H7 and H14)

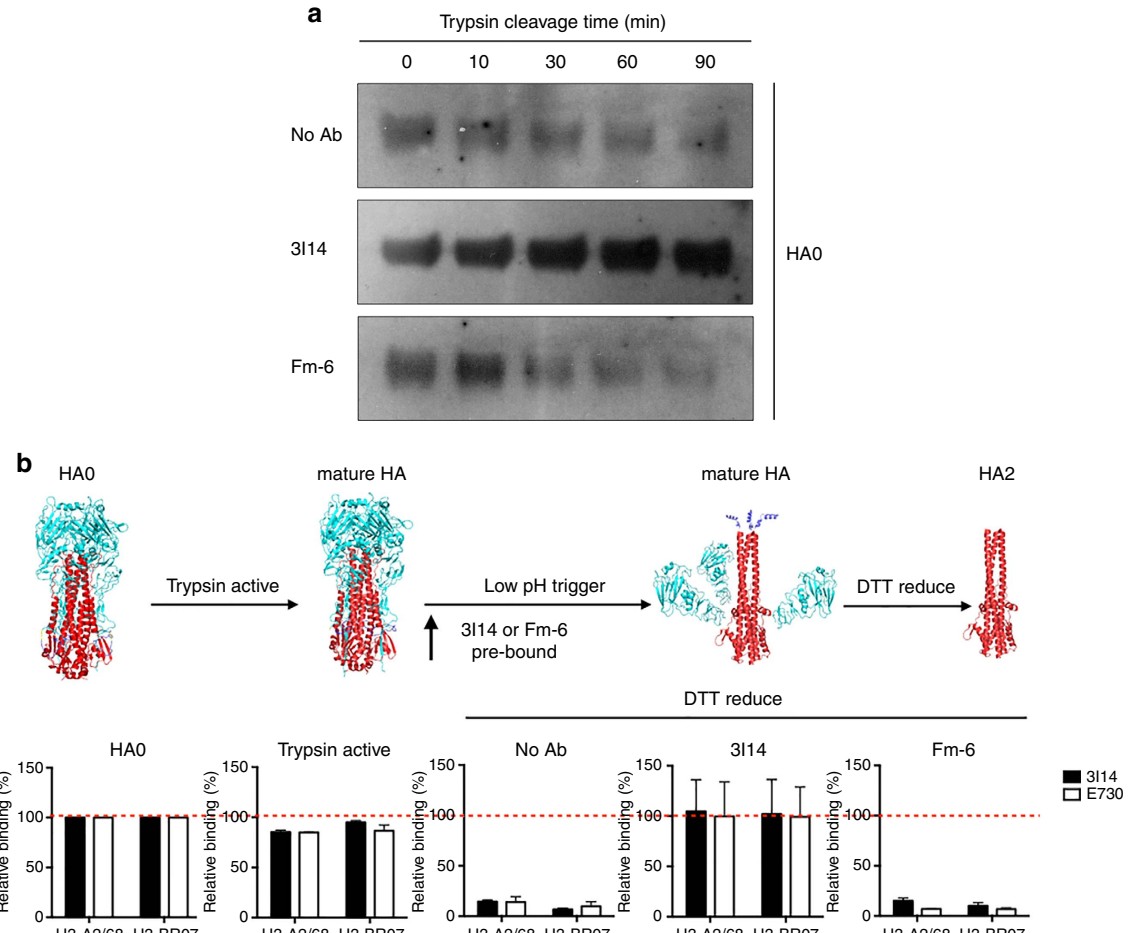

**Figure 4 | 3I14 blocks trypsin-mediated HA maturation and pH-dependent conformational changes.** (**a**) Trypsin Cleavage Inhibition Assay. 0.4 μg recombinant H3-histidine (H3-BR07) was incubated in the presence of 2.5 μg 3I14 or Fm-6 IgG1, or in the absence of antibody in Tris-HCl buffer at pH 8.0 containing 2 μg ml$^{-1}$ Trypsin at 37 °C. Trypsin digestion was stopped at several time-points by boiling the sample in a 100 °C water bath. Samples were run on 10% reduced SDS–PAGE and blotted using a HisProbe-HRP Abs. Data represent a representative experiment from three independent experiments. (**b**) 3I14 IgG1 prevented by low-pH triggered conformational rearrangements on the surface-expressed H3-A2/68 and H3-BR07. Upper panels show four various conformations of HA: uncleaved precursor (HA0, left); trypsin in neutral pH cleaved (mature HA, left middle); fusion pH cleaved (mature HA, right middle) and trimeric HA2 (HA2, right). The conformation rearrangements of surface-expressed H3 were detected by FACS staining of 3I14 (solid bars) and the head binding control mAb E730 (open bars). Binding is expressed as the percentage of binding to untreated HA (HA0). For this antibody inhibition assay, H3 was pretreated without mAb, with 3I14, or with control Ab, Fm-6 IgG1 before exposure of the cleaved HAs to pH 4.9. Data represent mean + s.d. of three independent experiments. SDS–PAGE, SDS–polyacrylamide electrophoresis.

and group 1 (H1, H5 and H9) with equilibrium binding constant, ($K_d$) ranging from 0.01 to 10 nM (Fig. 1c and Supplementary Fig. 3). 3I14 bound to all tested group 2 HAs (H3, H4, H7 and H14) with high affinity (mean $K_d < 0.1$ nM). In addition, 3I14 bound to group 1 H1 subtypes (H1-CA09, H1-SI06 and H1-PR8) with high affinity, whereas its affinity for other group 1 subtypes (H5-VN04, H5-IN05 and H9-HK99) was lower (mean $K_d = 1.02$, 1.05 and 5.23 nM, respectively).

3I14 potently neutralized numerous group 2 (H3 and H7) viruses including two reassortant viral strains (A/Wisconsin/67/05 (HA, NA) x A/Puerto Rico/8/34 and A/Aichi/2/68 (HA, NA) x A/Puerto Rico/8/34) and the novel H7N9 (A/Anhui/1/2013) strain with half maximal inhibitory concentration (IC$_{50}$) values ranging from 0.032 to 1.074 μg ml$^{-1}$ (Fig. 1d and Supplementary Fig. 4). It also neutralized pseudoviruses H7N1-FPN and H7N1-NL219 strains with IC$_{50}$ values ranging from 0.007 to 0.027 μg ml$^{-1}$ (Fig. 1e and Supplementary Fig. 4). In addition, 3I14 neutralized group 1 H1 strains (H1-CA09 and H1-PR8) with IC$_{50}$ values of 0.225 and 0.413 μg ml$^{-1}$ (Fig. 1d and Supplementary Fig. 4) and

pseudoviruses H5-VN04 and H5-HK97 with IC$_{50}$ values of 0.040 and 0.008 μg ml$^{-1}$, respectively (Fig. 1d and Supplementary Fig. 4).

**Epitope mapping and binding competition.** To investigate the epitope of HA for 3I14 recognition, we assessed its binding activity to either full-length of HA or HA1 subunit in Octet RED96 instrument. 3I14 bound trimeric full-length H3 strain, A/Perth/16/09 (PE-09), but did not bind its HA1 subunit (Supplementary Fig. 5). We further performed the binding competition assay between 3I14 and other stem-directed bnAbs: FI6v3, CR9114, 39.29, F10 and CR8020 (Fig. 2). 3I14 Fab strongly inhibits the binding of other anti-stem Abs CR9114, FI6v3 and 39.29 to H3-BR07 but not with the head-directed, anti-H3 mAb E730 (unpublished antibody sequence) (Fig. 2a–d). 3I14 also competes with CR8020, which is directed against a more membrane-proximal epitope[14]. 3I14 partially inhibits the binding of 39.29 and F10 to H5-VN04 but does not inhibit the binding of the anti-H5 head antibody 2A (ref. 12) (Fig. 2e,f). These results demonstrate that 3I14 is overlapping with or very close to the

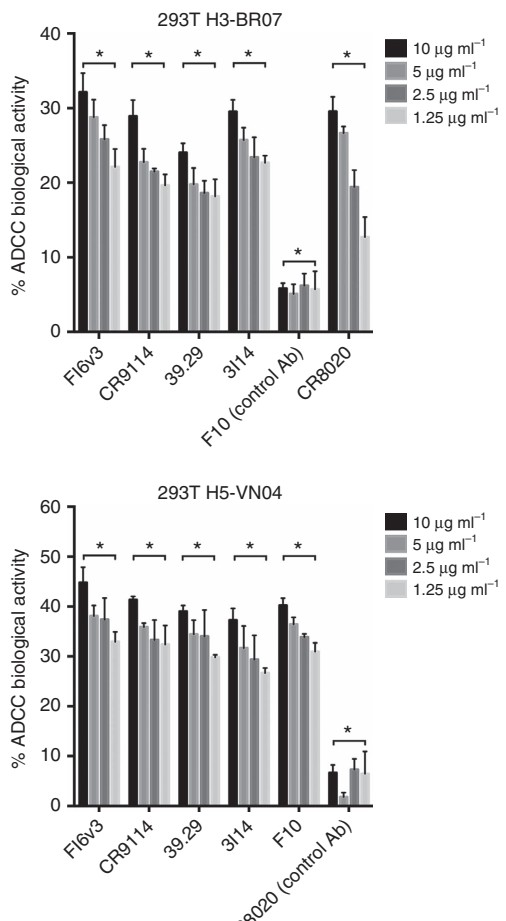

**Figure 5 | 3I14 mediates ADCC.** 3I14 and other anti-stem bnAbs, FI6v3, CR9114, 39.29, F10 and CR8020 induced antibody-dependent cellular cytotoxicity (ADCC) in H3- and H5-expressed 293T cells. Upper panel, $2 \times 10^4$/well H3-expressed 293T cells were attached to the plates before assay, and the medium was then replaced with 0.5% FBS low IgG serum assay buffer. Different bnAbs at concentration of 10, 5, 2.5 and 1.25 µg ml$^{-1}$ were added to each well. After 1 h, PBMCs were added at $1.2 \times 10^5$ cells/ well to assay plates and incubated for 6 h. The supernatants were harvested and detected using LDH cytotoxic kit by ELISA. Lower panel, H5-expressed 293T cells were seeded as target cells. Bars represent mean ± s.e.m. *P* value was calculated using two-way ANOVA, compared with F10 (upper) and CR8020 (lower). '*' represents *P* value for each comparison <0.0001. Data represent a representative experiment from three independent experiments. ANOVA, analysis of variance.

known stem epitopes of other bnAbs. In addition, 3I14 potently inhibited other bnAbs binding to H3 and moderately inhibited binding to H5. These results are consistent with the affinity measurements of 3I14 binding to H3 and H5.

**Prophylactic efficacy against influenza viruses in mice.** Next, we converted 3I14 into full-length human IgG1 to evaluate protective efficacy in a BALB/c mouse infection model against lethal strains that were available at the time of the study including H5N1, H3N2, H7N7 and H7N9 (Fig. 3). An anti-group 1 Ab, F10[12] was used as a strain-specific control. Mice were treated with varying doses of 3I14 and F10 IgG1 1 day before challenge with a lethal dose of H7N7-NL219, H7N9-AH13, H3N2-BR07-ma and H5N1-VN04 viruses. Prophylaxis using 5 mg kg$^{-1}$ 3I14 IgG1 fully protected mice from death after H7N7-NL219 or H7N9-AH13 challenge with minimal body weight loss at 14–18 days (Fig. 3a). At the dose of 25 mg kg$^{-1}$ 3I14 IgG1 showed 80% protection against H3N2-BR07 and 60% protection against H5N1-VN04. All surviving mice showed the reversal of weight loss by the end of the observation period (Fig. 3b).

**3I14 blocks HA maturation and conformational changes.** Stem-directed bnAbs are known to interfere with pH-dependent conformational changes and membrane fusion of HA[12,14,16]. Cleavage of the precursor HA0 primes HA for subsequent activation of membrane fusion in the acidic endosome environment. Immature HA0 is normally processed by surface protease on respiratory epithelial cells to HA1 and HA2 (refs 30,31), which is mimicked experimentally by treatment of HA0 with trypsin[32]. Since 3I14 targets the stem domain of HA comprising the HA0 cleavage site and the HA2 N-terminal fusion peptide, we tested whether 3I14 could also block trypsin cleavage activation of HA0 or interfere with HA-mediated virus-host membrane fusion. Figure 4a shows that 3I14 IgG1 but not control anti-SARS IgG1 (Fm-6) prevented cleavage of immature HA0. We also analysed 3I14's prevention of low-pH-triggered conformational rearrangements using the surface-expressed H3-A2/68 and H3-BR07. Figure 4b (upper) illustrates that 3I14 binds to both uncleaved HA precursor (HA0) (left) and two mature forms (HA), either after trypsin activation alone (left middle) or when followed by low-pH trigger (right middle). In contrast, it did not bind to dissociated HA2 mediated by DTT reduction (right). As expected, when 3I14 is pre-bound to mature HAs before low-pH trigger, the antibody maintained binding after DTT treatment (Fig. 4b, fourth panel), indicating that 3I14 inhibits the pH-dependent HA rearrangement (Fig. 4b, lower). In addition, pre-binding of 3I14 apparently prevented HA1-HA2 dissociation, because binding of E730 Ab (anti-HA1) was

**Table 2 | The binding affinity of 3I14 germline variants.**

| 3I14 Variants | H5-VN04 | | | H3-PE09 | | | H1-CA09 | | |
|---|---|---|---|---|---|---|---|---|---|
| | $K_d$ (nM) | $K_{on}$ (M$^{-1}$s$^{-1}$) | $K_{off}$ (s$^{-1}$) | $K_d$ (nM) | $K_{on}$ (M$^{-1}$s$^{-1}$) | $K_{off}$ (s$^{-1}$) | $K_d$ (nM) | $K_{on}$ (M$^{-1}$s$^{-1}$) | $K_{off}$ (s$^{-1}$) |
| GL | n* | n* | n* | 4.02 (−Δ15.3) | 1.39E+05 | 5.56E−04 (−Δ13.9) | 0.0597 (Δ4.7) | 1.09E+05 | 6.50E−06 (Δ7.5) |
| mHgL | 7.71 (−Δ7.5) | 2.58E+05 | 1.99E−03 (−Δ5.2) | 0.658 (−Δ2.5) | 1.92E+05 | 1.26E−04 (−Δ3.2) | <0.001† | 1.38E+05 | <1.0E−07† |
| gHmL | 1.95 (−Δ1.9) | 4.34E+05 | 8.44E−04 (−Δ2.2) | 0.733 (−Δ2.7) | 1.71E+05 | 1.25E−04 (−Δ3.1) | <0.001† | 1.55E+05 | <1.0E−07† |
| 3I14 WT | 1.02 | 3.75E+05 | 3.83E−04 | 0.263 | 1.52E+05 | 3.99E−05 | 0.279 | 1.74E+05 | 4.87E−05 |

WT, wild type.
n* indicates no binding detected.
†indicates no detectable dissociation.
(Δ) indicates the fold increase or (−Δ) fold decrease compared with WT.

preserved after DTT treatment (Fig. 4b lower). From these data we conclude that 3I14 binding to the HA stem epitope leads to inhibition of HA0 cleavage and pH-dependent conformational changes.

**3I14 mediates Fc-dependent cytotoxicity *in vitro*.** Anti-stem bnAbs are reported to efficiently mediate FcγR-dependent

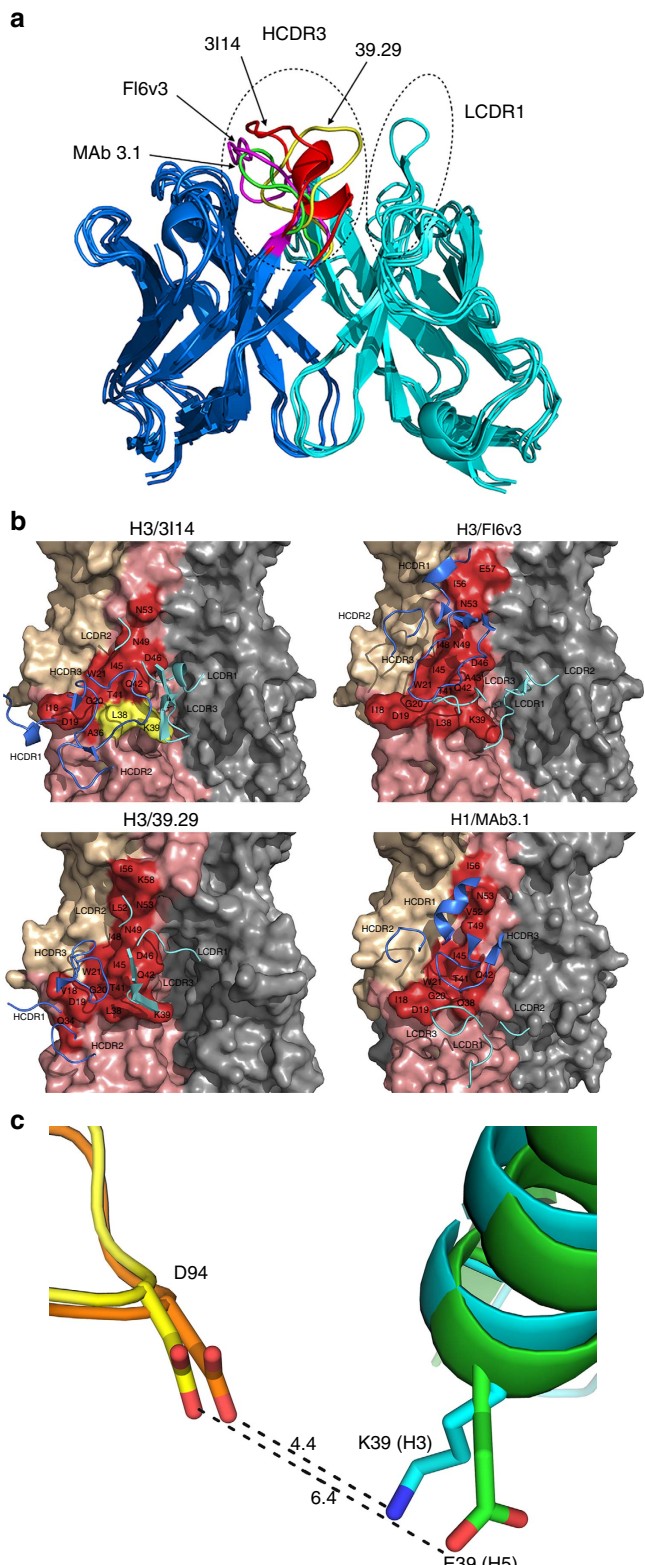

cytotoxicity of influenza virus-infected cells[33], which is considered to be a major mechanism of mAb-mediated antiviral clearance. To investigate the properties of antibody-dependent cellular cytotoxicity (ADCC) by 3I14 and other anti-stem bnAbs, we performed PBMC-based ADCC assay (Fig. 5) and a surrogate reporter-based assay (Supplementary Fig. 6) using HA-expressing 293T cells as targets. Following incubation with H3-expressing 293T target cells, 3I14 induced a significant lactose dehydrogenase (LDH) cytotoxicity response and luciferase signals in a dose-dependent manner and at a comparable level to other anti-stem bnAbs including FI6v3, CR9114, 39.29 and group 2 mAb CR8020 (Fig. 5 and Supplementary Fig. 6, upper panels). The specificity of this assay was demonstrated by the lack of response from anti-group 1 mAb, F10. 3I14 also specifically induced the ADCC activity against H5-expressed 293T cells mediated by human PBMCs, which can reach 40% of total cell lysis. A group control, CR8020 did not have shown this property (Fig. 5, lower panel). We also observed 3I14 induced luciferase reactivity against H5-expressed 293T target cells (Supplementary Fig. 6, lower panel). These data support that 3I14 engages an Fc-dependent immune-mediated mechanism similar to other anti-stem bnAbs[33].

**Somatic hypermutations *in vivo*.** To assess the contribution of SHMs on affinity maturation, we produced the 3I14 VH and VL germline versions (3I14-GL), and chimeric antibodies formed by mature (m) 3I14 heavy chain paired with germline (g) light chain (3I14-mHgL) and vice versa (3I14-gHmL) (Supplementary Fig. 2). The 3I14 variant antibodies were expressed as human IgG1 and their binding affinity against H1, H5 and H3 was evaluated (Table 2 and Supplementary Fig. 7). Remarkably, 3I14-GL variant still bound H3 and H1 in the nM and sub-nM range while showing a > 15-fold decrease in binding affinity to H3 and a 4.7-fold increase in binding affinity to H1, respectively (Table 2). These changes in 3I14-GL binding affinity to H3 and H1 were predominantly caused by an increase and decrease in dissociation rate constant ($K_{off}$) by 13.9- or 7.5-fold, respectively. Interesting, 3I14-GL did not bind H5 under these assay conditions.

Comparing the two chimeric forms to wild-type (WT) 3I14, the SHMs present in both VH and VL of 3I14 appear to make equal contributions to H3 binding ($K_d$: 0.658 nM versus 0.733 nM). In addition, both the heavy and light chain chimeras resulted in essentially irreversible binding to H1 with $K_{off} < 1.0E-7 \text{ s}^{-1}$. However, in the case of H5, VL mutations contribute more

**Figure 6 | Modelling of 3I14 and docking with H3/H5.** (**a**) The superimposition of 3I14 model with FI6v3, 39.29 and MAb 3.1. The bnAbs are displayed in ribbon representations. The heavy chain is in blue and the light chain is in cyan. The HCDR3s and LCDR1s are indicated by the hatched ovals. The residues in HCDR3 of 3I14, FI6v3, 39.29 and MAb 3.1 are coloured in red, magenta, yellow and green, respectively. (**b**) The complex structures of *IGHV3-30*-derived bnAbs with HAs. The epitope residues on the HAs are displayed in surface representation and the CDR loops of bnAbs are shown are shown as ribbons. HA1 of HA monomer is in wheat, HA2 is in salmon and epitope residues are labelled as red. The key residues L38 and K39 are coloured in yellow. Heavy chain CDRs of bnAbs are in blue and light-chain CDRs are in cyan. 3I14 was homology modelled using the antibody-modelling module in BioLuminate and the model was superimposed to H3/39.29 complex structure before docking with RosettaDock. Other three *IGHV3-30* bnAbs, FI6v3, 39.29 and MAb 3.1 were downloaded from Protein Data Bank. (**c**) The interaction of D94 in 3I14 with H3/H5. H3 is shown in cyan with K39 shown as stick; H5 is shown in green with E39 shown as stick; 3I14 is shown in orange in H3/3I14 model and in yellow in H5/3I14 model with D94 shown as stick.

**Table 3 | The binding affinity of 3I14 VLD94N variants.**

| 3I14 Variants | H5-VN04 | | | H3-PE09 | | |
|---|---|---|---|---|---|---|
| | $K_d$ (nM) | $K_{on}$ (M$^{-1}$s$^{-1}$) | $K_{off}$ (s$^{-1}$) | $K_d$ (nM) | $K_{on}$ (M$^{-1}$s$^{-1}$) | $K_{off}$ (s$^{-1}$) |
| 3I14 WT | 1.02 | 3.27E+05 | 3.87E−04 | 0.263 | 1.52E+05 | 3.99E−05 |
| 3I14 VLD94N | 0.187 | 3.83E+05 | 7.74E−05 | 0.308 | 1.77E+05 | 5.44E−05 |

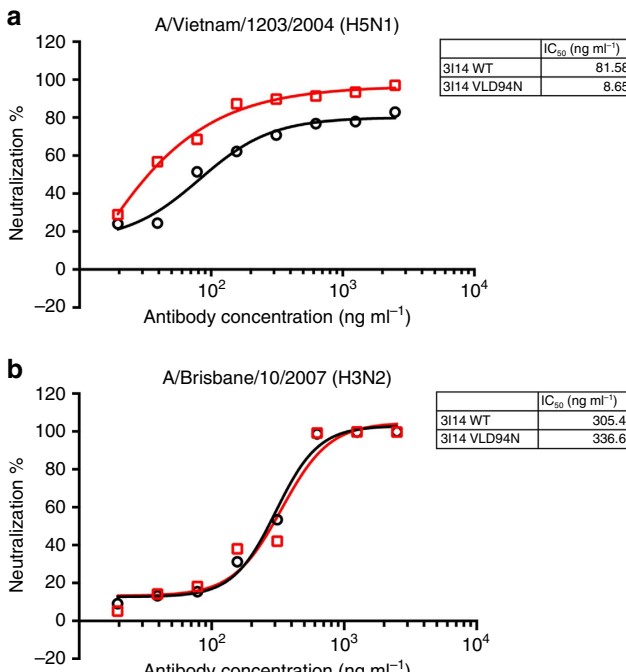

**Figure 7 | 3I14 and VLD94N variant neutralized HA pseudotyped virus H5N1-VN04 and infectious virus H3N2-BR07.** The 3I14 (black) and VLD94N variant (red) neutralized pseudotyped virus H5N1-VN04 (**a**) and H3N2-BR07 virus (**b**). These data represent average neutralization titres of two to three independent experiments.

to the affinity increase than VH mutations (7.5-fold versus 1.9-fold) due to a 5.2-fold and 2.2-fold decrease in $K_{off}$, respectively. From these studies we conclude that 3I14-GL shows higher affinity binding to H1 and moderate affinity to H3 with changes in $K_{off}$ being largely responsible for kinetic differences to compare to 3I14-WT. For H5 binding, the SHMs in 3I14 are absolutely required for binding with VL mutations providing a greater contribution to binding than VH changes. All changes in binding affinity to H1, H3 and H5 are mainly the consequence of changes in dissociation rate ($K_{off}$) constants.

**Structural based affinity maturation *in vitro*.** To characterize the molecular basis for unequal binding strengths to H3 and H5 and to engineer 3I14 with improved affinity to H5N1 strains, we first used an antibody structure prediction program BioLuminate[34] for *in silico* simulation of the 3I14 structure. The superimposition of the 3I14 model with other three *IGHV3-30* bnAbs, FI6v3, 39.29 and MAb 3.1, is shown in Fig. 6a. It is clear that the major difference among these antibodies is the conformation of HCDR3 with the exception of the longer LCDR1 of FI6v3 that forms a loop structure which makes contact with HA.

Next, the 3I14 model was docked to the H3 trimer structure with RosettaDock server[35]. Since 3I14 competes with FI6v3 and 39.29 for binding to H3 and H5, and MAb 3.1 occupies the same

conserved epitope with FI6v3 and 39.29 (ref. 18), we hypothesized that 3I14 adopts a similar binding scheme to interact with H3/H5 as FI6v3, 39.29 and MAb 3.1. For these three Ab-HA co-crystal structures, HCDR3 plays a major role in forming a hydrophobic core with the fusion peptide and helix A[15,17,18]. Rather than making significant interactions with HA, HCDR1 and HCDR2 appear to stabilize the HCDR3 loop to facilitate binding. The hydrophilic light chain CDR residues also interact with HA and surround the hydrophobic core; however, the orientation of the light chains are not conserved nor are the residues involved in binding. These observations suggest that the light chains mainly contribute to the binding by orienting the HCDR3 to an optimal position to interact with the epitope.

On the basis of these solved co-crystal structures, we chose from 1000 decoys the most similar binding models of 3I14/H3 (Fig. 6b) and 3I14/H5 complexes (not shown). A thorough analysis of the interfaces of the two complexes was carried out in order to understand why 3I14 binds H3/H1 stronger than H5 (Supplementary Table 1). Energetic calculations[36] show very favourable binding contributions between D94 of 3I14 light chain and K39 in the H3 model, which may form a salt bridge while E39 is rotating away from D94 in the H5/3I14 model due to the electrical repulsion and may be unfavourable for H5 binding (Fig. 6c). In addition to H5, the E39 amino acid change is also found in group 1 H2, H6, H11 and H13 strains (Supplementary Table 2). Another striking variation is at H3 position L38, where this residue is changed to K/R38 for some group 1 strains (Supplementary Table 2). However, the binding contribution shows L/K38 contacts HCDR3 residues Y104, F105 and F109 in both models with favourable to very favourable binding (~70% of the total favourable free energy) and therefore we considered these residues to have a positive effect on binding to both HAs.

**VLD94N variant improves the binding and neutralization to H5.** Next, to eliminate the proposed repulsive effect of E39 and D94, we hypothesized a single Asp-to-Asn (D to N) mutation that leads to a loss of a negative charge at the site will bind equally well to both H3 and H5. To examine this structural based modification, we first evaluated the binding affinity of both WT 3I14 and VLD94N variant IgG1. As shown in Table 3, VLD94N variant increased binding affinity to H5 by nearly 10-fold but did not cause any significant change in binding to H3. Interestingly, the higher affinity to H5 was also due to decreased dissociation rates, while association rates were equal (Table 3 and Supplementary Fig. 8).

We also performed neutralization assays to assess the activity of 3I14 VLD94N variant against H5 pseudotyped or H3 infectious virus (Fig. 7). Compared with 3I14, the VLD94N variant neutralized H5-VN04 pseudovirus with 10-fold higher potency (IC$_{50}$: 8.65 ng ml$^{-1}$ versus 81.58 ng ml$^{-1}$) (Fig. 7a). Meanwhile, the neutralization activity against H3-BR07 remained intact of VLD94N variant (IC$_{50}$: 336.6 ng ml$^{-1}$ versus 305.4 ng ml$^{-1}$) (Fig. 7b). These results demonstrate that the optimized 3I14 VLD94N variant lead to an increase in binding and neutralizing ability to H5 while maintaining its efficacy to H3.

## Discussion

B-cell memory is a second layer of humoral immune defense and is essential for the survival of the host. If pre-existing neutralizing antibodies (nAbs) secreted by long-lived plasma cells are insufficient to bind and eliminate the invading microbe, pathogen-specific memory B cells will rapidly undergo pro-liferation and differentiation to produce nAbs[24]. Compared with long-lived plasma cells, memory B cells provide broader antibody responses through affinity maturation. In this study, we analysed the cross-reactivity of 237 H3-reactive memory B cells from seven healthy donors and found that 20 (8.44%) also bound H7 and H1 HAs. This H3/H7/H1 hetero-subtypic population corresponds to 46.5% (43 clones) of H3/H7 binding and 64.5% (31 clones) of H3/H1 binding populations (Table 1). This broad reactivity represents an unexpectedly high level of hetero-subtypic group 2 and groups 2/1 influenza A binding clones than might be expected based on published plasmablast cell studies[37,38]. These memory B-cell frequencies are also particularly remarkable since the blood samples were not harvested immediately post seasonal vaccination. Other memory B-cell clones including H3-reactive B cells that cross bind to influenza B, while found at lower levels (3.38%), represent an important finding that requires further evaluation as they may represent an important population to elicit with universal influenza vaccines.

3I14 mAb is the first high-affinity bnAb to be isolated from human memory B cells that has activity against groups 1 and 2 influenza A viruses. Two other IGHV3-30-encoded bnAbs with both group 1 and 2 activities—FI6 and 39.29—were isolated from plasma cells and plasmablasts, respectively[15,17]. These three examples provide evidence of biased use of a second VH germline gene in the generation of bnAbs beyond IGHV1-69 for which such preferential use has now clearly been demonstrated[29,39,40]. The IGHV3-30 germline gene has 19 alleles and we are able to make tentative assignments of 39.29 (ref. 17) and MAb 3.1 (ref. 18) to IGHV3-30*01, while FI6 (ref. 15) and 3I14 to IGHV3-30*03. There are a number of single-nucleotide polymorphisms in the coding region of the IGHV3-30 alleles that result in amino acid changes but only one CDR residue at position 33 in HCDR1 has A/G variance among the alleles. On the basis of three crystal structures and our 3I14 modelling, position 33 does not appear to be directly involved in HA binding. Whether variable levels of IGHV3-30 Abs exist in the expressed Ab repertoires of individuals that encode different IGHV3-30 alleles due to variances in the noncoding regions that may affect transcription/recombination will require additional investigations to answer.

There are notable similarities and differences among the stem-directed bnAbs encoded by IGHV1-69 and IGHV3-30 in their structures, V segment SHMs[39] (Supplementary Table 3) and involvement of light chains in binding. The majority of stem-directed IGHV1-69 bnAbs have broad hetero-subtypic binding and neutralization activity against group 1 influenza A strains[29,39], although two IGHV1-69 BnAbs- CR9114 (ref. 16) and MAb 1.12 (ref. 19) -generated from human-derived phage display libraries have broader activity with groups 1 and 2 activity. For their heavy chains, several studies have shown that IGHV1-69 bnAbs require two anchor residues: F54 in HCDR2 and a properly positioned tyrosine in HCDR3, as well as a hydrophobic amino acid at position 53 and a minimal number of signature mutations in CDR1/2 to generate high-affinity antibodies[29,39,40]. In contrast to IGHV1-69-encoded antibodies, IGHV3-30 bnAbs utilize HCDR3 to form a hydrophobic core that contributes to HA binding[15,17,18] and possess a longer HCDR3 due to a large amount of N-insertions in the VDJ junctions[15,18,29]. For IGHV1-69-encoded bnAbs, F10, CR6261, CR9114 and MAb 1.12, the light chains can be replaced without affinity impairments[12,13,19,40]. A distinguishing feature of 3I14 as

well as the other IGHV3-30 bnAbs is that the light chains are important for binding both when in germline configuration and as shown with 3I14 when SHMs are introduced which leads to broadening of its antigenic breadth (Table 2). Specifically, the 3I14 chiral S31 in VL germline LCDR1 likely applies tension to the binding interface and sequentially changes the coordinates of the main chain (Supplementary Fig. 2), which may lead to a failure to provide flexibility for conformational changes on HA binding that could result in fast dissociation rates (Table 2). As shown in Supplementary Fig. 9, this liability is overcome by the G31 substitution in WT 3I14, which leads to strong interactions with Q42 and D46 of H5. Other mutations in the light chain are buried in the interior of the antibody and may not contribute to the binding to HA. Further, our structure-based modification, D94N in LCDR3, also provided the same binding improvement towards H5 (Table 3). N94 can compensate for the electrostatic repulsive force between E39 in H5 and D94 in 3I14, facilitating the assembly of H5-3I14 complex. This D94N mutation could be generated in somatic B cells in vivo by activation-induced deamination (AID) U:G mismatch repair[41]. By constructing an antibody mutant through single site mutagenesis, we show that a point mutation in LCDR3 is sufficient to increase H5 binding affinity by mimicking the similar molecular mechanisms of SHM.

3I14 shares the conserved HA stem epitope with FI6v3 and 39.29 bnAbs and shows similar mechanisms of action in inhibiting HA maturation, membrane fusion, and inducing ADCC of virally infected cells. Unlike FI6v3 and 39.29, however, 3I14 mAb shows lower binding and neutralizing activities to some group 1 viral subtypes, such as H5. The 3I14 germline reversion lacks binding affinity to H5 but retains its strong reactivity with H3/H1, suggesting the 3I14 clone was selected by high-affinity binding to H3 and/or H1 and the accumulation of SHMs conferred broader neutralizing activity against the H5 subtypes. Therefore, we speculate that in this H5N1 naive donor stochastic rather than an induced process resulted in the acquisition of H5 binding. This observation also builds on classic studies that the diversity of memory BCRs increases by accumulating selectable and unselectable mutations during the antigen recalling responses[42].

In summary, the isolation and characterization of 3I14 mAb suggests that memory B cells may continuously undergo intraclonal diversification via accumulation of SHMs to broaden antibody reactivity against divergent influenza viruses. Interestingly, 3I14-like memory B cells could significantly enhance H5 reactivity through SHMs, while at the same time, reduce their inherent higher affinity for H1. To maintain a broader bnAb repertoire against rapidly diversifying seasonal viruses or potentially pandemic viruses, 3I14-like memory B cells are a general pool that we wish to preserve and expand in that they are not terminally committed to any HA subtype; rather they maintain a plasticity potential that can be evoked by minimal additional SHMs. This is supported by our back-mutation studies that show 3I14 GL antibody has broad activity against H1 and H3 subtypes and has the potential to gain broader activity against H5. This is a distinguishing feature from the GL version of plasma-derived FI6 and other bnAbs which greatly loose activity against some subtypes while maintaining strong activity against other subtypes[15,29]. We propose an improved vaccination strategy that preferentially targets the HA stem may maintain the longevity of 3I14-like and other anti-stem memory B cells. For example, sequential vaccination with chimeric HAs (cHA) may allow these subdominant memory B cells to preferentially expand[43,44]. Likewise, direct focusing of Ab repertoires on the HA stem domain by vaccination with headless HAs may achieve a similar goal[45,46]. Additionally, targeting influenza A vaccines to a naive

pool of *IGHV3-30* B cells that provide strong initial binding energy through their BCR H/L CDR3 motifs may prove useful. These vaccine approaches may have the advantage of preventing the memory B cell loss that is further compounded by HA antigen induced expansion of abundant anti-head memory B cell clones[47,48]. A different strategy may be to inhibit the mammalian target of rapamycin (mTOR) pathway which has been reported to modify the nAb repertoire and increase the frequency of Abs potentially cross-reactive to conserved stem epitopes[49,50]. Only future research on the biological, molecular and genetic composition of HA-specific memory B cells and their evolution will reveal how to properly elicit bnAbs in the development of 'universal' influenza vaccines[51].

## Methods

**Cells.** Fresh PBMCs were isolated by processing seven deidentified, discarded leukoreduction collars from platelet apheresis that are left for research purposes at the Kraft Family Blood Donor Center, DFCI and Specimen Bank, Brigham & Woman's Hospital (BWH). The donors all consented for the research use of the discarded/deidentified collars at time of donation. The use of these human blood samples for research purpose was approved under IRB protocols from Partners (BWH Protocol #2005-P-001364/3) and DFCI Protocol #14-343. Madin–Darby canine kidney (MDCK) cells, 293T and 293 F cells were obtained from American Type Culture collection (Manassas, VA, USA).

**Preparation of recombinant haemagglutinins.** The extracellular domain of H3 (A/Brisbane/10/2007), residues 17–531, was expressed as fusion protein included a C-terminal peptide containing Avitag (amino acid sequence: GGGLNDIFEAQKIEWHE), thrombin cleavage site, trimerization T4 fibritin foldon domain and six histidine residues. The fusion protein H3-ATTH was expressed in 293 F cells and purified from the supernatant by Ni-NTA affinity chromatography. Purified recombinant HA protein was cleaved by thrombin enzyme (Novagen, Darmstadt, Germany), then biotinylated with BirA enzyme (Avidity, Aurora, CO) according to the manufacturer's instructions.

The full-length HA genes of A/New York/18/09 (H1-NY09), A/Texas/05/09 (H1-TX09), A/Japan/305/57 (H2-JP57), A/Aichi/2/68 (H3-A2/68), A/Brisbane/10/07 (H3-BR07), A/Netherlands/2/2005 (H4-NL05), A/Vietnam/1203/04 (H5-VN04), A/Hong Kong/156/97 (H5-HK97), A/chicken/New York/14677-13/98 (H6-NY98), A/Netherlands/219/03 (H7-NL219), A/turkey/Ontario/6118/68 (H8-ON68), A/Hong Kong/1073/99 (H9-HK99), A/duck/Memphis/546/74 (H11-MEM74), A/duck/Alberta/60/76 (H12-AB76), A/mallard/Astrakhan/263/1982 (H14-AS82), A/shearwater/West Australia/2576/79 (H15-WA79) and A/black-headed gull/Sweden/2/99 (H16-SE06) were cloned into pcDNA3.1 vector and transfected into 293T/17 cells to produce cell surface-expressed HA.

Recombinant full-length HA proteins of H1 subtypes A/California/04/09 (H1-CA09), A/Solomon Islands/3/06 (H1-SI06) and A/Puerto Rico/8/34 (H1-PR8); H3 A/Perth/16/09 (H3-PE09), A/Uruguay/716/07 (H3-UY07), and A/Victoria/341/11 (H3-VIC11); H5 A/Vietnam/1203/04 (H5-VN04) and A/Indonesia/05/05 (H5-ID05); H7 A/Netherlands/219/03 (H7-NL219), A/Canada/RV444/04 (H7-CA444) and A/Anhui/1/13 (H7-AH13); H9 A/Hong Kong/1073/99 (H9-HK99) were obtained from the NIH BEIR Repository (NIH, Manassas, VA). Recombinant full-length HAs of subtypes H4 A/mallard/Netherlands/2/05 (H4-NL05) and H14 A/mallard/Astrakhan/263/82 (H14-AS82) were kindly gifted from Dr R. C. Liddington (Burnham Institute for Medical Research, CA, USA).

**Preparation of influenza viruses and HA pseudotyped viruses.** Wild-type influenza viruses A/California/4/09 (H1N1-CA09), A/Puerto Rico/8/34 (H1N1-PR8), A/Perth/16/09 (H3N2-PE09), A/Aichi/2/68 (H3N2-A2/68), A/Hong Kong/8/68 (H3N2-HK68), A/Sydney/5/97 (H3N2-SY97), A/Brisbane/10/07 (H3N2-BR07), A/Wisconsin/67/05 (HA, NA) × A/Puerto Rico/8/34 (H3N2), A/Aichi/2/68 (HA, NA) × A/Puerto Rico/8/34 (H3N2) and A/Nanchang/993/95 (H3N2-NC95) were obtained from the NIH BEIR Repository (NIH, Manassas, VA), and grown in MDCK cells by standard viral culture techniques. A/Brisbane/10/2007-ma (H3N2) used in animal challenge studies is a mouse-adapted virus derived from a PR8 reassortant virus × -171 (ref. 52).

The full-length HA genes of A/Vietnam/1203/04 (H5-VN04), A/Hong Kong/156/97 (H5-HK97), A/Netherlands/219/07 (H7-NL219), A/FPV/Rostock/1934 (H7-FPV) and neuramidase gene N1 of H5-VN04 (Genbank accession AAW80723) were cloned into pcDNA3.1 plasmids, separately. The Env-Pseudotyped luciferase reporter viruses were produced in 293 T/17 cells by co-transfection with four plasmids: HA-expressing plasmid, N1-expressing plasmid pcDNA3.1-N1, HIV packaging vector pCMVΔR 8.2 encoding HIV-1 Gag-Pol and transfer vector pHIV-Luc encoding the firefly luciferase reporter gene control of the HIV-1 LTR[12]. Briefly, the pcDNA3.1-H5-VN04, H5-HK97, H7-NL219 or H7-FPV plasmids were separately co-transfected into 293T/17 cells with the N1-expressing plasmid pcDNA3.1-N1-VN04, HIV packaging vector pCMVΔR 8.2

and reporter vector pHIV-Luc. The ratio of HA- to N1-expressing plasmids was 4:1. Viral supernatants were harvested at 48 h post transfection. Viral titration was evaluated by measuring luciferase activity using the POLARstar Omega Microplate Reader (BMG LABTECH, Ortenberg, Germany).

**FACS sorting of H3 binding memory B cells.** Fresh PBMCs were isolated from the collected blood by use of the Ficoll-Paque gradient (GE HealthCare). The CD19[+]/CD27[+] B cells were stained with biotinylated H3-ATTH and allophycocyanin (APC)-labelled streptavidin. Single H3-reactive memory B cells were excluded doublets using SSC and FSC gate (Supplementary Fig. 10), then were sorted into 384-well plate. After 14 days of expansion, the supernatants were tested for reactivity to recombinant H1 (H1-CA09), H3 (H3-BR07) and H7 (H7-CA444) HA proteins and were analysed by the Meso Scale Discovery multiplex (MSD, Rockville, Maryland). Subsequently, the reactive supernatants were measured *in vitro* neutralizing activity against H3N2-BR07. All H3N2 neutralizing antibodies were rescued by single-cell RT–PCR using primers as previously described[53].

**Expression and purification of 3I14 scFv and IgG antibodies.** We used a rapid single-step cloning procedure to initially move the 3I14 Ab into the pcDNA3.1-Hinge scFvFc minibody expression vector, generating the scFv as a fusion product with the hinge, $CH_2$, and $CH_3$ domains of human IgG1 (ref. 12). Purified 3I14 scFvFc was used to assess the binding and neutralizing activity against multiple HAs and viruses of different subtypes (Fig. 1). For whole-human IgG1s, the gene fragments of the scFv were separately subcloned into human IgG1 expression vector TCAE6 (ref. 54). The scFvFcs or IgG1s were expressed in 293F cells by transient transfection and purified by protein A sepharose affinity chromatography.

**Kinetic and $K_d$ determinations.** Kinetic analyses of bnAbs binding to recombinant HAs were performed on biolayer interferometry using an Octet RED96 instrument (ForteBio, Menlo Park, CA) at 25 °C. The bnAbs at 5 nM were captured onto anti-human IgG Fc biosensors in Pierce protein-free blocking buffer (PBS with 0.5% (v/v) Tween-20) for 180 s. Recombinant full-length HAs were loaded at concentrations ranging from 6.25 to 100 nM. All experiments contained an additional anti-human IgG Fc antibody biosensor that tested for potential nonspecific interactions between HAs and anti-human IgG Fc. For the measurement of the association rate constant ($K_{on}$), association of 3I14 was measured for 300 or 600 s by exposing the sensors to up to 20 concentrations of HAs. For the measurement of $K_{off}$, dissociation of 3I14 IgG1 was measured for 900 or 1200 sc. The $K_d$ values were calculated using ForteBio Data Analysis 7.0 software. All binding traces and curves used for fitting are reported in Supplementary Fig. 3.

**Microneutralization assay.** Before the experiment, MDCK cells ($1.5 \times 10^4$ cells per well) were seeded to the 96-well tissue culture plates and washed twice with PBS, then incubated in DMEM media supplemented with 2 µg ml$^{-1}$ trypsin and 0.5% BSA. One hundred TCID$_{50}$ (median tissue culture infectious doses) of virus were mixed in equal volume with two fold serial dilutions of Ab or antibody containing supernatant in 96-well plates, and incubated for 1 h at 37 °C. After the 1 h incubation, the Ab-virus mixture was transferred to confluent MDCK monolayers in duplicate, followed by incubation at 37 °C for 21 h. After 21 h infection, Ab-virus mixture was moved and cells were washed with PBS, fixed in 80% acetone, and viral antigen detected by indirect ELISA with a mAb against influenza A Virus Nucleoprotein (NP) (clone A3, BEI).

**Prophylactic studies in mice.** Twenty-four hours before virus challenge inoculation groups of five female 8–10 weeks old BALB/c mice (Charles River, Wilmington, MA) were injected with 3I14 and F10 IgG1 at low dose (5 mg kg$^{-1}$) or high dose (25 mg kg$^{-1}$) by intraperitoneal (i.p.) route in 0.5 ml volume, respectively. All groups of mice were intranasally infected 10 LD$_{50}$ (median lethal dose) of A /Vietnam/1203/04 (H5N1), A/Brisbane/10/07-ma (H3N2), A/Netherlands/219/03 (H7N7) or A/Anhui/1/13 (H7N9). Mice were weighed on the day of virus challenge and then monitored for clinical signs and body weight recorded daily for 14 or 18 days. Body weight loss of ≥ 25% relative to initial weight, or a score of 4 on clinical signs (unresponsiveness or severe neurological symptoms such as hind limb paralysis, ataxia) index were used as survival endpoints. The animal research described in this study was approved by the Institutional Animal care and Use Committee (IACUC) of the Centers for Disease Control and Prevention (CDC). The CDC animal facility is accredited by the Association for Assessment and Accreditation of Laboratory Animal Care International.

**Antibody binding competition.** A final concentration of 5 µg ml$^{-1}$ H3-BR07 or H5-VN04 protein immobilized on ELISA plates were incubated with two fold serial dilution of 3I14 Fab, ranging from 80 to 0.3 nM and mixed with other scFvFc Abs at 5 nM. After co-incubation for 1 h, the binding of scFvFc Abs was detected using horseradish peroxidase-conjugated anti-human $CH_2$ antibodies (Life Technologies, Grand Island, NY) and measured using Super AquaBlue ELISA substrate (ebioscience, San Diego, CA) on the POLARstar Omega Microplate Reader (BMG LABTECH, Ortenberg, Germany).

**Trypsin cleavage inhibition assay.** A total of 0.4 µg recombinant H3-histidine (H3-ATTH) protein was incubated in the presence of 2.5 µg 3I14 or anti-SARS Fm-6 IgG1, or in the absence of antibody in Tris-HCl buffer at pH 8.0 containing 2 µg ml$^{-1}$ Trypsin-ultra (New England Biolabs, Ipswich, MA) at 37 °C. Trypsin digestion was stopped at several time-points by boiling the sample in a 100 °C water bath. Samples were run on 10% SDS–polyacrylamide electrophoresis gel under reducing conditions and blotted using a HisProbe-horseradish peroxidase and SuperSignal West HisProbe Kit (Pierce Biotechnology, Rockford, IL). Images have been cropped for presentation in Fig. 4a. Full-size images are presented in Supplementary Fig. 11. Data represent a representative experiment from three independent experiments.

**Conformational change flow cytometry assay.** 293T/17 cells were transfected with full-length recombinant influenza A pcDNA3.1-H3-A2/68 and H3-BR07 plasmids. At ~30 h after transfection, cells were detached from the culture vessel using 0.2% ethylenediaminetetraacetic acid (EDTA). To measure mAb binding to different HA structural conformations, cell samples were exposed to different treatments, aliquoted and stained with 3I14 or E730 scFvFc Abs. Detached cells consecutively treated with 2 µg ml$^{-1}$ trypsin (Gibco, Grand Island, NY) for 5 min at room temperature, washed with 1% BSA/PBS and incubated for 15 min in citric acid-sodium phosphate buffer pH 4.9, washed, and then incubated for 20 min with 50 mM dithiothreitol (DTT) in PBS at room temperature. Alternatively, 5 µg 3I14 or anti-SARS Ab Fm-6 IgG1 was added before the low-pH step. Samples of consecutive treatments were stained with APC-conjugated anti-human Fc (BioLegend, San Diego, CA). Stained cells were analysed using a BD FACSAria II with FACS Diva software (Becton Dickinson, Franklin Lakes, NY).

**Antibody-dependent cell cytotoxicity assay (ADCC).** The ADCC assay was performed on HAs-expressed 293T cells as with fresh PBMCs from healthy human donors. The ADCC activity was determined by a LDH release assay (Pierce Biotechnology, Rockford, IL). Fresh PBMCs (effector cells) were isolated from the collected blood by use of the Ficoll-Paque gradient (GE HealthCare). As target cells, $2 \times 10^4$/well H3- or H5- expressed 293T cells were attached to the solid round bottom 96-well plates before assay, and the medium was then replaced with low IgG Serum assay buffer (RPMI 1640 with 0.5% low IgG FBS). The scFvFc antibodies were added to each well at 10, 5, 2.5 and 1.25 µg ml$^{-1}$ final concentration. After one-hour, PBMCs were added for $1.2 \times 10^5$/well to assay plates in Low IgG Serum assay buffer and incubated for 6 h. The supernatants were recovered by centrifugation at 300g and measured using LDH Cytotoxicity Assay Kit (Pierce Biotechnology, Rockford, IL) at 490 and 680 nm by the Benchmark Plus Reader (Bio-Rad, Hercules, CA). The LDH activity was determined by subtracting the 680 nm absorbance value (background) from the 490 nm absorbance reading. The per cent cytotoxicity was calculated as: %Cytotoxicity = 100 × (E−s.e.−ST)/(M−ST); E, released LDH from E/T culture with antibody; s.e., spontaneous released LDH from effectors; ST, spontaneous released LDH from targets; M, the maximum released LDH from lysed targets. Data represent a representative experiment from three independent experiments, and all tests were performed in triplicate.

**Surrogate reporter-based ADCC assay.** The ADCC Reporter Bioassay uses engineered Jurkat cells stably expressing the FcγRIIIa receptor, V158 (high-affinity) variant, and an NFAT response element driving expression of firefly luciferase as effector cells[55] (Promega). Antibody biological activity in ADCC is quantified through the luciferase produced as a result of NFAT pathway activation; luciferase activity in the effector cell is quantified with luminescence readout. As target cells, $1 \times 10^4$/well H3- or H5- expressed 293T cells were attached to the flat bottom 96-well plates before assay, and the medium was then replaced with Low IgG Serum assay buffer (RPMI 1640 with 0.5% low IgG FBS). scFvFc antibodies were added to each well at 5, 1, 0.2 and 0.04 µg ml$^{-1}$ final concentration. After one-hour, Jurkat effector cells were added for $6.0 \times 10^4$/well to assay plates in low IgG Serum assay buffer and incubated for 6 h. The supernatants were recovered by centrifugation at 300g and measured luminescence using Bio-Glo Luciferase Assay kits (Promega, Madison, WI) by the POLARstar Omega Microplate Reader (BMG LABTECH, Ortenberg, Germany). Data represent a representative experiment from three independent experiments, and all tests were performed in triplicate.

**Sequence analysis.** The full-length influenza A HA sequences were downloaded from the Influenza Virus Resource at the National Center for Biotechnology Information (NCBI) database. The Phylogenetic (PHYML) trees are based on their amino acid sequence comparison using Geneious software. The new bnAb, 3I14, was rescued by single-cell RT–PCR and was analysed for germline gene usage, SHMs, N-nucleotides insertion and cognate variable heavy (VH) and light (VL) chain gene pairs using IMGT database (http://imgt.cines.fr). Antibody variants in which single or multiple germline mutations were reverted to the germline were produced by synthesis (Genewiz, South Plainfield, NJ) and confirmed by sequencing. The VH and VL sequences of F10 (3FKU), FI6v3 (3ZTJ), CR9114 (4FQY), CR8020 (3SDY) and 39.29 (4KVN) were obtained through the Protein Data Bank and the corresponding genes were synthetized and expressed by transient transfection.

**In silico structure modelling.** 3I14 was homology modelled using the antibody-modelling module as implemented in BioLuminate (Schrödinger, Inc). Briefly, the heavy chain and light chain sequence were entered into the program, and the template for framework (FRH/FRL region) was identified by searching against antibody structure database. The PDB file 4GXU was selected as suitable template due to the best composite score (0.94) considering both heavy and light chains. Then the CDR loops-simulated 3I14 model was generated after analysing all available antibody loop structures. The model was further superimposed to H3/39.29 complex structure before docking with RosettaDock. Only high resolution docking is performed with side chain and loop rearrangement allowed. 1000 decoys were generated for each docking and clustered based on RMSD values. The final model was selected based on the cluster size and the criteria described in the result session.

**Statistical analyses.** Data was analysed using two-way analysis of variance and Dunnett's multiple comparisons test. '*' indicate $P < 0.0001$ of each comparison. All values and bars are represented as mean ± s.d.

**Data availability.** Sequence data that support the findings of this study have been deposited in GenBank with the primary accession codes KX588237 and KX588238. The full-length influenza A HA sequences were downloaded from the Influenza Virus Resource at the National Center for Biotechnology Information (NCBI) database. The sequences used in Fig. 1a for analysis were: H1 (A/New York/18/2009, GenBank: GQ457506.1), H2 (A/Japan/305/1957, GenBank: AY643087.1), H3 (A/Brisbane/10/2007, GenBank: KM978061.1), H4 (A/mallard/Netherlands/2/2005, GenBank: GU052580.1), H5 (A/Vietnam/1203/2004, GenBank: HM006759.1), H6 (A/chicken/New York/14677-13/1998, GenBank: CY014880.1), H7 (A/Netherlands/219/03, GenBank: AY338459.1), H8 (A/turkey/Ontario/6118/1968, GenBank: CY130046.1), H9 (A/Hong Kong/1073/99, GenBank: AJ404626.1), H10 (A/mallard/Sweden/51/2002, GenBank: HM136575.1), H11 (A/duck/Memphis/546/1974, GenBank: CY130070.1), H12 (A/duck/Alberta/60/1976, GenBank: CY130078.1), H13 (A/gull/Maryland/704/1977, GenBank: CY130086.1), H14 (A/mallard/Astrakhan/263/1982, GenBank: CY130094.1), H15 (A/shearwater/Australia/2576/1979, GenBank: CY130102.1) and H16 (A/shorebird/Delaware/172/2006, GenBank: CY130110.1). The authors declare that all other data supporting the findings of this study are available within the article and its Supplementary Information Files.

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

## Acknowledgements

We thank Drs De-Kuan Chang and Xian-Chun Tang for technical support. This work was supported by National Institutes of Health (NIAID U01-AI074518) to W.A.M. The findings and conclusions in this report are those of the authors and do not necessarily represent the views of the Centers for Disease Control and Prevention or the Agency for Toxic Substances and Disease Registry.

## Author contributions

Y.F. designed and performed all *in vitro* study. Y.F. and Z.Z. carried out epitope mapping, antibodies modelling and protein–protein docking. Y.F., Z.Z., Ja.S. and T.S. purified antibodies. Y.F. and Y.A. analysed the attribution of binding energy in antigen–antibody complexes. Y.F. and Ji.S. irradiated the cells. Y.F., Z.Z. and W.A.M. analysed data and drafted the manuscript. Ja.S. edited the manuscript. C.R., M.J.H. and L.-M.C. performed animal study as well as neutralization assay with influenza virus H7N9-AH13. Y.F., Q.Z., R.O.D. and W.A.M. finalized the paper. All authors commented on the manuscript.

## Additional information

**Competing financial interests:** The authors declare no competing financial interests.

