## [Peer review file · Nature Communications]

Reviewers' Comments:

Reviewer #1 (Remarks to the Author)

Here Fu et al. report the characterization of a novel stem-directed, broadly neutralizing influenza antibody. They demonstrate that the clone, 3I14, can successfully bind and neutralize both group 1 and 2 influenza viruses by binding in the HA stem region and preventing maturation of the HA molecule. This antibody improved survival and reduced weight loss in mice infected with H7N7, H7N9, H3N2 and H5N1 virus subtypes. An in-vitro reporter assay suggests that 3I14 can interact with Fcγ receptor IIIa and activate FcR-bearing reporter cells in the presence of "target" cells expressing HA. This is consistent with a potential role for 3I14 in mediating ADCC, which has been suggested to be the mechanism by which influenza bnAbs provide protection in vivo. The investigators show through sequencing of BCRs from the cloned memory B cells and germline cells that the heavy chain of the 3I14 antibody is encoded by IGHV3-30, that the CDR3 region includes a relatively long loop, and that the antibody has undergone an intermediate level of somatic mutation that involves both heavy and light chains. Finally, they use crystal structures of HA with modeled 3I14 to show that binding of HA likely involves both the heavy and light chains. Structural analysis also allows the investigators to predict residues that would increase H5N1 binding; indeed, they show that altering these residues results in an increase in binding affinity to H5N1 HAs.

This is a comprehensive study of a novel bnAb derived from memory B cells and provides important insights into the mechanisms by which bnAbs against influenza can be generated in vivo. Interestingly, the sequence analysis show that H5 HA binding was likely acquired during somatic hypermutation, underscoring the important role of this process in driving bnAb ontogeny.

A few minor revisions, including editing for clarity, grammar, and syntax, would improve this manuscript.

Minor points

1. The authors test the in-vivo protective efficacy of 3I14 against H5N1, H3N2, H7N7 and H7N9 viruses. It seems unusual that they chose to use 2 H7 viruses but not an H1, as a broadly neutralizing antibody with efficacy against H1 viruses would be highly valuable to the human population. The authors should justify their choice of viruses.

2. Several influenza bnAbs have been described, but generating such responses through vaccination has proven difficult, and it is not clear whether the results described here, interesting as they are, shed further light on how to design bnAb-inducing antigens. The authors somewhat elide this important point in both the abstract and discussion. In the abstract they state that "establishing an optimized memory B cell precursor pool" should be a goal of vaccination. But they do not define what this means, or provide suggestions for strategies that would do this. They approach the idea in discussion lines 388-389, mentioning "precisely engineered antigen variants" that have been described as an HIV-1 vaccine concept, but they do not define how these variants will be precisely engineered, either for HIV or influenza. In HIV, designing these variants relies on analyses of the co-evolution of viral envelope genes and antibody repertoires over time in infected individuals. It would be interesting if the authors could speculate in more detail about how this strategy might be applied to influenza.

3. Fig. 5 and associated methods: The "ADCC" assay reported by the authors measures engagement of FcR in the presence of HA and not ADCC effector activity per se. The authors should note this caveat. Also, what is the level of background luciferase activity in this assay? Is background subtracted from the reported results? What controls are used in these experiments?

4. Fig. 3: It is very difficult to tell the differences between the symbols, as they are all filled and of similar size. A combination of symbol sizes, and/or a combination of filled and open symbols would help visually distinguish the different antibodies and doses.

5. Fig 6a: It would be helpful to include legends in the figures explaining which antibody structures are represented by which colors.

Grammar/syntax/spelling

The manuscript as written suffers from multiple grammatical and syntactical errors that diminish clarity. Some examples are listed below.

1. Line 62 "a highly conserved epitopes" should read "...epitope"

2. Line 65 the sentence beginning "Whereas..." is a fragment

3. Lines 138-156: References to Fig. 1 panels are all off by one letter (1a should be 1b, etc)

4. Lines 170-171 "3I14 is a potent inhibitor of H3..." Does this mean that 3I14 neutralizes H3 and H5 viruses, or that it inhibits binding of other antibodies to HA of these subtypes?

5. Lines 176-177 should read "AN anti-group 1 Ab, F10, was USED as A STRAIN-specific control."

6. Lines 312-314: I am not sure what this sentence is saying. "...heterosubtypic H3/H1/H7 binding which corresponds to 46.5% of H3/H7 populations..." If I interpret the sentence and the table correctly, it would be clearer to restate something like this: "We found 237 memory B cell clones from 7 subjects that bound H3 HA. 43 of these also bound H7 HA. Of these 43, 20 clones (46.5%) made antibodies that also bound H1 HA."

Reviewer #2 (Remarks to the Author)

"A broadly neutralizing anti-influenza antibody reveals ongoing capacity of hemagglutinin-specific memory B cells to evolve" presents an overview regarding VH3-30-derived broadly neutralizing antibody 3I14. The authors demonstrate that memory B cell evolution can broaden the breadth of neutralizing antibodies. Fu et al. also show that a D94N amino acid substitution in the variable light chain CDR1 improves binding and neutralization activity to the H5 HA subtype.

Concerns with this manuscript are as follows:

- Fu et al. state "Antigen-specific human memory B cells (CD19+CD27+) were isolated from peripheral blood mononuclear cells of 7 healthy donors using tetramerized H3 trimers...". With the use of the tetramerized H3 trimers, how do the authors rule out the possibility of two memory B cells binding the same "bait", which would be viewed as a single event during the single-cell sorting? A representative FACS plot for all of the gating schemes should be provided.
- The authors find that mAb 3I14 "has 15 variable heavy chain and 7 variable light chain somatic mutations...". How do the authors explain lack of extensive somatic hypermutation? Broadly neutralizing antibodies that target conserved epitopes on the hemagglutinin protein of influenza A viruses tend to have extensive somatic hypermutation on the heavy chain (CR9114, CR6261, CR8020, S6-B01, 2D04, etc. Is this due to the light chain assisting in antigen binding? This issue should be addressed.
- When compared to other broadly neutralizing influenza-specific antibodies, mAb 3I14 has relatively weak potency in the murine animal model (80% survival at 25 mg/kg). How do the authors explain that an antibody targeting an epitope "shared" with previously described broadly neutralizing antibodies behaves so poorly?
- The authors claim that mAb 3I14 mediates Fc-dependent viral clearance, but fail to provide

compelling data to support this hypothesis. Firstly, there is no evidence that mAb 3I14 allows for virus clearance. Lung titers should be assessed to prove that administration of this mAb lowers virus titers. Secondly, ADCC is only one mechanism of Fc-dependent engagement. The authors should include a positive control (CR9114, etc) in this assay and provide fold-induction results for the Promega kit, as described by the manufacturer. If the authors want to clearly prove that this mAb relies on Fc-dependent viral clearance, they should clone the variable regions into a mouse IgG2a backbone and a mouse IgG2a D265A backbones (Dillilo et al. 2012). This would allow for the direct comparison of a functional Fc region and a dysfunctional Fc region (abrogates Fc engagement).

- The methodology for reverting back to germline needs to be described in more detail. Sup. Fig. 2 shows that 3I14 GL is based on VH3-30*18, IGHD3-22*01 and IGHJ4*02. However, R108S, F109G, V110Y, W111Y and V112Y are not included in 3I14 GL. This needs to be addressed.

Additional concerns with this manuscript are as follows:

- Fu et al. state, "We converted 3I14 into full-length human IgG1". The authors should mention which isotype mAb 3I14 was originally.

- Other group 1/group 2 cross-reactive antibodies (S6-B01, 2D04, etc) have been described (Henry Dunand et al. 2015). Appropriate references should be included.

- Would this VLD94N mutation also increase binding and neutralization against non-pseudoviruses? Or improve potency in vivo?

- The authors need to provide more information regarding the microneutralization assay performed. Is the antibody also included in the replenishing media after infection?

This manuscript provides an overall summary regarding broadly neutralizing mAb 3I14. Major and minor concerns should be addressed before the manuscript is considered for publication.

Reviewers' comments:

Reviewer #1 (Remarks to the Author):

Here Fu et al. report the characterization of a novel stem-directed, broadly neutralizing influenza antibody. They demonstrate that the clone, 3I14, can successfully bind and neutralize both group 1 and 2 influenza viruses by binding in the HA stem region and preventing maturation of the HA molecule. This antibody improved survival and reduced weight loss in mice infected with H7N7, H7N9, H3N2 and H5N1 virus subtypes. An in-vitro reporter assay suggests that 3I14 can interact with Fcγ receptor IIIa and activate FcR-bearing reporter cells in the presence of "target" cells expressing HA. This is consistent with a potential role for 3I14 in mediating ADCC, which has been suggested to be the mechanism by which influenza bnAbs provide protection in vivo. The investigators show through sequencing of BCRs from the cloned memory B cells and germline cells that the heavy chain of the 3I14 antibody is encoded by IGHV3-30, that the CDR3 region includes a relatively long loop, and that the antibody has undergone an intermediate level of somatic mutation that involves both heavy and light chains. Finally, they use crystal structures of HA with modeled 3I14 to show that binding of HA likely involves both the heavy and light chains. Structural analysis also allows the investigators to predict residues that would increase H5N1 binding; indeed, they show that altering these residues results in an increase in binding affinity to H5N1 HAs.

This is a comprehensive study of a novel bnAb derived from memory B cells and provides important insights into the mechanisms by which bnAbs against influenza can be generated in vivo. Interestingly, the sequence analysis show that H5 HA binding was likely acquired during somatic hypermutation, underscoring the important role of this process in driving bnAb ontogeny.

A few minor revisions, including editing for clarity, grammar, and syntax, would improve this manuscript.

Minor points

1. The authors test the in-vivo protective efficacy of 3I14 against H5N1, H3N2, H7N7 and H7N9 viruses. It seems unusual that they chose to use 2 H7 viruses but not an H1, as a broadly neutralizing antibody with efficacy against H1 viruses would be highly valuable to the human population. The authors should justify their choice of viruses.

A: We chose H3N2 as the strain for memory B cells baiting, screening and micro-neutralization and we had an in vivo passaged strain that was lethal in the mouse model. Group 1 H5N1 and Group 2 H7N7 viruses are both high pathogenic strains and were lethal in the mouse model. H7N9 emerged in the human population in the spring of 2013 when these animal studies were being performed, which was lethal in the mouse model and was tested as well. We did not have a lethal strain of H1N1 available at the time that these animal experiments were performed.

2. Several influenza bnAbs have been described, but generating such responses through vaccination has proven difficult, and it is not clear whether the results described here, interesting as they are, shed further light on how to design bnAb-inducing antigens. The authors somewhat elide this important point in both the abstract and discussion. In the abstract they state that "establishing an optimized memory B cell precursor pool"

should be a goal of vaccination. But they do not define what this means, or provide suggestions for strategies that would do this. They approach the idea in discussion lines 388-389, mentioning "precisely engineered antigen variants" that have been described as an HIV-1 vaccine concept, but they do not define how these variants will be precisely engineered, either for HIV or influenza. In HIV, designing these variants relies on analyses of the co-evolution of viral envelope genes and antibody repertoires over time in infected individuals. It would be interesting if the authors could speculate in more detail about how this strategy might be applied to influenza.

A: The discussion section was revised and more details and references are included (lines 389-395 in revised manuscript).

Line 389-395, "For example, sequential vaccination with chimeric HAs (cHA) may allow these subdominant memory B cells to preferentially expand^{1,2}. Likewise, direct focusing of Ab repertoires on the HA stem domain by vaccination with headless HAs may achieve a similar goal^{3,4}." We also mention use of inhibitors of mTOR pathway that was been tested in vivo^{5,6}.

3. Fig. 5 and associated methods: The "ADCC" assay reported by the authors measures engagement of FcR in the presence of HA and not ADCC effector activity per se. The authors should note this caveat. Also, what is the level of background luciferase activity in this assay? Is background subtracted from the reported results? What controls are used in these experiments?

A: We performed new PBMC-based LDH cytotoxicity assays to directly document Fc-dependent ADCC (new Figure 5) in addition to the surrogate reporter-based ADCC assay (now Supplementary Fig. 6). As shown in Fig. 5, 3I14 specifically induced the LDH cytotoxicity against H5- or H3-expressed target cells mediated by human PBMCs, which can reach 40% or 30% of total cell lysis, respectively. The bnAbs, CR9114, FI6v3 and 39.29 were used as positive controls, whereas F10 and CR8020 were used as groups-specific controls. Significant differences between the 3I14 and groups-specific controls Abs were shown.

*The measurement details are shown in **Methods: Antibody-dependent cell cytotoxicity assay (ADCC)**. The LDH activity was determined by subtracting the 680 nm absorbance value (background) from the 490 nm absorbance reading. The percent cytotoxicity was calculated as: %Cytotoxicity = $100 \times (E - SE - ST) / (M - ST)$; E, released LDH from E/T culture with antibody; SE, spontaneous released LDH from effectors; ST, spontaneous released LDH from targets; M, the maximum released LDH from lysed targets.*

4. Fig. 3: It is very difficult to tell the differences between the symbols, as they are all filled and of similar size. A combination of symbol sizes, and/or a combination of filled and open symbols would help visually distinguish the different antibodies and doses.

A: We edited Fig. 3 using colored and filled/open symbols combinations to distinguish Abs and dose.

5. Fig 6a: It would be helpful to include legends in the figures explaining which antibody structures are represented by which colors.

A: We labeled the HCDR3 and LCDR1 by the hatched ovals, and indicated the HCDR3 of different bnAbs in Fig. 6a.

Grammar/syntax/spelling

The manuscript as written suffers from multiple grammatical and syntactical errors that diminish clarity. Some examples are listed below.

1. Line 62 "a highly conserved epitopes" should read "...epitope"

A: Corrected.

2. Line 65 the sentence beginning "Whereas..." is a fragment

A: Corrected.

3. Lines 138-156: References to Fig. 1 panels are all off by one letter (1a should be 1b, etc)

A: Corrected throughout.

4. Lines 170-171 "3I14 is a potent inhibitor of H3..." Does this mean that 3I14 neutralizes H3 and H5 viruses, or that it inhibits binding of other antibodies to HA of these subtypes?

A: The sentence is corrected.

Line 171-172, "In addition, 3I14 potentially inhibited other bnAbs binding to H3 and moderately inhibited binding to H5."

5. Lines 176-177 should read "AN anti-group 1 Ab, F10, was USED as A STRAIN-specific control."

A: Corrected.

6. Lines 312-314: I am not sure what this sentence is saying. "...heterosubtypic H3/H1/H7 binding which corresponds to 46.5% of H3/H7 populations...?" If I interpret the sentence and the table correctly, it would be clearer to restate something like this: "We found 237 memory B cell clones from 7 subjects that bound H3 HA. 43 of these also bound H7 HA. Of these 43, 20 clones (46.5%) made antibodies that also bound H1 HA."

A: We corrected this sentence.

Line 312-315, "...found that 20 of them (8.44%) also bound H7 and H1 HAs. The H3/H7/H1 heterosubtypic population corresponds to 46.5% (of 43 clones) of H3/H7 binding and 64.5% (of 31 clones) of H3/H1 binding populations (Table 1)."

Reviewer #2 (Remarks to the Author):

"A broadly neutralizing anti-influenza antibody reveals ongoing capacity of hemagglutinin-specific memory B cells to evolve" presents an overview regarding VH3-30-derived broadly neutralizing antibody 3I14. The authors demonstrate that memory B cell evolution can broaden the breadth of neutralizing antibodies. Fu et al. also show that a D94N amino acid substitution in the variable light chain CDR1 improves binding and neutralization activity to the H5 HA subtype.

Concerns with this manuscript are as follows:

- Fu et al. state "Antigen-specific human memory B cells (CD19+CD27+) were isolated from peripheral blood mononuclear cells of 7 healthy donors using tetramerized H3 trimers...". With the use of the tetramerized H3 trimers, how do the authors rule out the possibility of two memory B cells binding the same "bait", which would be viewed as a single event during the single-cell sorting? A representative FACS plot for all of the gating schemes should be provided.

A: We exclude doublets from Flow Cytometry sorting. A representative FACS plot for all of the gating schemes were provided below.

- The authors find that mAb 3I14 "has 15 variable heavy chain and 7 variable light chain somatic mutations...". How do the authors explain lack of extensive somatic hypermutation? Broadly neutralizing antibodies that target conserved epitopes on the hemagglutinin protein of influenza A viruses tend to have extensive somatic

hypermutation on the heavy chain (CR9114, CR6261, CR8020, S6-B01, 2D04, etc. Is this due to the light chain assisting in antigen binding? This issue should be addressed.

A: We have previously reported that IGHV1-69 based anti-stem sBnAbs (e.g. F10, CR9114, CR6261) have an average of 12.6 ± 4.2 VH segment SHM which is average for rearranged antibody genes regardless of target specificity (Avnir, 2014 PLoS. Path.)⁷. And their VL's are not involved in HA binding. MAb 3I14 has the same level of SHM in VH and even less in VL. We listed the SHMs in both heavy chain (VH) and light chain (VL) of 6 bnAbs (amino acids sequences were analyzed on IMGT online tool).

mAb	SHMs of Amino acid	
	VH (V+D+J)	VL (V+J)
3I14	15 (11+1+3)	7 (6+1)
FI6v3	7 (6+1+0)	10 (9+1)
CR9114	21 (17+3+1)	11 (11+0)
CR6261	19 (15+1+3)	7 (7+0)
CR8020	16 (13+2+1)	7 (7+0)
F10	15 (13+1+1)	5 (5+0)

For IGHV3-30 encoded antibodies, FI6v3 and 3I14 bound to HAs using both VH and VL. These IGHV3-30 bnAbs differ from IGHV1-69 bnAbs in that the former have longer HCDR3's due to a large number of insertions at the VDJ junctions (as we discussed in the paper).

- When compared to other broadly neutralizing influenza-specific antibodies, mAb 3I14 has relatively weak potency in the murine animal model (80% survival at 25 mg/kg). How do the authors explain that an antibody targeting an epitope "shared" with previously described broadly neutralizing antibodies behaves so poorly?

A: Here, we listed the mouse survival rates in published studies of different bnAbs during in vivo viral challenge. MAbs 39.29⁸ and CT149⁹ are both bnAbs that share the same epitope with FI6v3, CR9114 and 3I14. They show 60%-100% protection efficacy against different viral strains, respectively.

	H3N2	H5N1	H7N7	H7N9	H1N1
3I14 (25 mg/kg)	80%	60%	100%	100%	-
CT149 (30 mg/kg)	100%	100%	-	70%	100%
39.29 (15 mg/kg)	60%-100% *	-	-	-	90%
39.29 (45 mg/kg)	100%	-	-	-	100%

** The protective efficacy depends on different H3N2 strains. – indicates no testing results.*

The protection efficacy depends on the dose of bnAbs, the binding affinity against HAs and the pathogenicity of viruses. 3I14 potently bound to two H7 strains (H7-A/Netherlands/219/2003 and H7-A/Anhui/1/2013) with high affinity: (mean Kd = 0.67 and 0.0336 nM, respectively, Fig. 1c) and provided 100% protection. But its binding affinity to H5-VN04 (A/Vietnam/1203/2004) was lower (Kd= 1.02 nM, Fig. 1c). The in vivo viral challenge result is consistent with the affinity measurements of 3I14 in vitro.

- The authors claim that mAb 3I14 mediates Fc-dependent viral clearance, but fail to provide compelling data to support this hypothesis. Firstly, there is no evidence that mAb 3I14 allows for virus clearance. Lung titers should be assessed to prove that administration of this mAb lowers virus titers.

*A: We changed the subtitle of the **Results** section from “3I14 mediates Fc-dependent viral clearance” to “3I14 mediates Fc-dependent cytotoxicity in vitro”. We also changed the final sentence from “...3I14 also likely engages an Fc-dependent immune-mediated mechanism for in vivo protection” to “...3I14 engages an Fc-dependent immune-mediated mechanism similar to other anti-stem bnAbs¹⁰” and do not claim 3I14 mediates in vivo viral clearance. Lung titers were not assessed in our animal studies.*

Secondly, ADCC is only one mechanism of Fc-dependent engagement. The authors should include a positive control (CR9114, etc) in this assay and provide fold-induction results for the Promega kit, as described by the manufacturer.

A: We performed new PBMC-based LDH cytotoxicity assay to measure the ADCC activity (new Figure 5) in addition to the surrogate reporter-based ADCC assay (now Supplementary Fig. 6). As shown in Fig. 5, 3I14 specifically induced the LDH cytotoxicity against H5- or H3-expressed target cells mediated by human PBMCs, which can reach 40% or 30% of total cell lysis, respectively. The bnAbs, CR9114, FI6v3 and 39.29 were used as positive controls, whereas F10 and CR8020 were used as groups-specific controls. These data suggested that 3I14 also engages an Fc-dependent immune-mediated mechanism like CR9114, FI6v3 and 39.29.

If the authors want to clearly prove that this mAb relies on Fc-dependent viral clearance, they should clone the variable regions into a mouse IgG2a backbone and a mouse IgG2a D265A backbones (Dillilo et al. 2012). This would allow for the direct comparison of a functional Fc region and a dysfunctional Fc region (abrogates Fc engagement).

A: We would like to thank the reviewer for this suggestion. We hope to perform the suggested study that 3I14 mediates Fc-dependent viral clearance in vivo in our future work.

- The methodology for reverting back to germline needs to be described in more detail. Sup. Fig. 2 shows that 3I14 GL is based on VH3-30*18, IGHD3-22*01 and IGHJ4*02. However, R108S, F109G, V110Y, W111Y and V112Y are not included in 3I14 GL. This needs to be addressed.

A: Based on the nucleotide analysis of 3I14 on IMGT online tool^{11, 12}, these 5 amino acids represent “N” additions that occur at the D_HJ_H junction and are not encoded by either D_H or J_H germline gene.

Additional concerns with this manuscript are as follows:

- Fu et al. state, “We converted 3I14 into full-length human IgG1”. The authors should mention which isotype mAb 3I14 was originally.

A: The mAb 3I14 is generated from human IgG secreting cells (data not shown). 3I14 sequence was derived from RT-PCR and firstly expressed as a single chain antibody. Then we convert 3I14 into IgG1 format to perform in vivo viral challenge. Here we mentioned “human IgG1” to distinguish single chain format of 3I14.

- Other group 1/group 2 cross-reactive antibodies (S6-B01, 2D04, etc) have been described (Henry Dunand et al. 2015). Appropriate references should be included.

*A: Dunand et al. published 3 Abs, named 045-051310-2B06, 042-100809-2F04, and S6-B01¹³. Only MAbs 2B06 and S6-B01 bound to H1 and H5 (S6-B01 bound to H1 with low binding activity) whereas 2F04 only bound to group 2 HAs. There is not 2D04 mAb mentioned. Only 2B06 neutralized the H1N1 (A/California/04/2009) strain in plaque reduction assays, whereas S6-B01 did not. No in vivo viral challenge result against H1N1 in this paper. We included 2B06 as a bnAb and the Dunand 2015 reference in the **Introduction**.*

- Would this VLD94N mutation also increase binding and neutralization against non-pseudoviruses? Or improve potency in vivo?

A: VLD94N variant increases binding to H5 strain (A/Vietnam/1203/04). We do not have any isogenic matched pseudovirus and WT H5N1 influenza strain to perform this comparison study. Previous published studies have demonstrated the neutralization dose-response curves for anti-influenza nAbs against pseudoviruses and corresponding viruses were similar^{14, 15, 16, 17}. Therefore we believe that VLD94N mutant will also show increased neutralization to infectious H5N1 both in vitro and in vivo.

- The authors need to provide more information regarding the microneutralization assay performed. Is the antibody also included in the replenishing media after infection?

*A: We add the description in **Method: “Microneutralization assay”***

Line 496-497, “After 21h infection, Ab-virus mixture were moved...”

This manuscript provides an overall summary regarding broadly neutralizing mAb 3I14. Major and minor concerns should be addressed before the manuscript is considered for publication.

References:

1. Krammer F, Pica N, Hai R, Margine I, Palese P. Chimeric hemagglutinin influenza virus vaccine constructs elicit broadly protective stalk-specific antibodies. *J Virol* **87**, 6542-6550 (2013).
2. Nachbagauer R, et al. Induction of broadly reactive anti-hemagglutinin stalk antibodies by an H5N1 vaccine in humans. *J Virol* **88**, 13260-13268 (2014).
3. Yassine HM, et al. Hemagglutinin-stem nanoparticles generate heterosubtypic influenza protection. *Nature medicine* **21**, 1065-1070 (2015).
4. Impagliazzo A, et al. A stable trimeric influenza hemagglutinin stem as a broadly protective immunogen. *Science* **349**, 1301-1306 (2015).
5. Mannick JB, et al. mTOR inhibition improves immune function in the elderly.

Science translational medicine **6**, 268ra179 (2014).

6. Keating R, *et al.* The kinase mTOR modulates the antibody response to provide cross-protective immunity to lethal infection with influenza virus. *Nature immunology* **14**, 1266-1276 (2013).
7. Avnir Y, *et al.* Molecular signatures of hemagglutinin stem-directed heterosubtypic human neutralizing antibodies against influenza A viruses. *PLoS pathogens* **10**, e1004103 (2014).
8. Nakamura G, *et al.* An in vivo human-plasmablast enrichment technique allows rapid identification of therapeutic influenza A antibodies. *Cell Host Microbe* **14**, 93-103 (2013).
9. Wu Y, *et al.* A potent broad-spectrum protective human monoclonal antibody crosslinking two haemagglutinin monomers of influenza A virus. *Nature communications* **6**, 7708 (2015).
10. DiLillo DJ, Tan GS, Palese P, Ravetch JV. Broadly neutralizing hemagglutinin stalk-specific antibodies require FcγR interactions for protection against influenza virus in vivo. *Nature medicine* **20**, 143-151 (2014).
11. Alamyar E, Duroux P, Lefranc MP, Giudicelli V. IMGT((R)) tools for the nucleotide analysis of immunoglobulin (IG) and T cell receptor (TR) V-(D)-J repertoires, polymorphisms, and IG mutations: IMGT/V-QUEST and IMGT/HighV-QUEST for NGS. *Methods Mol Biol* **882**, 569-604 (2012).
12. Giudicelli V, Lefranc MP. IMGT/junctionanalysis: IMGT standardized analysis of the V-J and V-D-J junctions of the rearranged immunoglobulins (IG) and T cell receptors (TR). *Cold Spring Harbor protocols* **2011**, 716-725 (2011).
13. Henry Dunand CJ, *et al.* Preexisting human antibodies neutralize recently emerged H7N9 influenza strains. *J Clin Invest* **125**, 1255-1268 (2015).
14. Tsai C, *et al.* Measurement of neutralizing antibody responses against H5N1 clades in immunized mice and ferrets using pseudotypes expressing influenza hemagglutinin and neuraminidase. *Vaccine* **27**, 6777-6790 (2009).
15. Zhang S, *et al.* Generation and characterization of an H5N1 avian influenza virus hemagglutinin glycoprotein pseudotyped lentivirus. *Journal of virological methods* **154**, 99-103 (2008).
16. Wang W, *et al.* Establishment of retroviral pseudotypes with influenza hemagglutinins from H1, H3, and H5 subtypes for sensitive and specific detection of neutralizing antibodies. *Journal of virological methods* **153**, 111-119 (2008).
17. Ding H, Tsai C, Zhou F, Buchy P, Deubel V, Zhou P. Heterosubtypic antibody response elicited with seasonal influenza vaccine correlates partial protection against highly pathogenic H5N1 virus. *PLoS One* **6**, e17821 (2011).

Reviewers' Comments:

Reviewer #1 (Remarks to the Author)

The authors have extensively revised the manuscript in response to reviewer comments. In particular they add a new analysis of ADCC activity that provides a much better surrogate of cell killing than the reporter assay used in the initial submission.

I have only a few minor remaining comments:

Minor points:

Several reviewer comments are addressed only in the rebuttal letter and not in the text or supplemental material. While I appreciate reading the authors' rationale for choosing the virus strains for in-vivo challenge experiments in the rebuttal, this information should be included in the manuscript itself so that readers of the paper can also understand. Several of reviewer 2's comments are similarly addressed only in rebuttal. For example, it would be helpful for the final manuscript to include information about the FACS gating strategy and exclusion of doublets, and also the comparison of levels of somatic hypermutation in 3I14 and other bnAbs.

The authors add a few sentences in the end of the discussion section summarizing various vaccination approaches that have been used to attempt to stimulate bnAbs against influenza. However, this still does not explain why they believe that characterization of 3I14 (or 3I14-expressing memory B cells) per se constitutes a conceptual advance toward the goal of "universal" influenza vaccination. Similarly, while they have slightly modified the statement in the abstract about an "optimized memory B cell pool," the revised discussion does not explain either the criteria by which memory B cell pools might be considered "optimized" or speculate on how specifically vaccine modalities might do this. The authors should add a few sentences placing 3I14 in context with other bnAbs. They should also either explain how 3I14-like cells represent "optimized" memory B cells or else modify this statement in the abstract.

Reviewer #2 (Remarks to the Author)

"A broadly neutralizing anti-influenza antibody reveals ongoing capacity of hemagglutinin-specific memory B cells to evolve" presents an overview regarding VH3-30-derived broadly neutralizing antibody 3I14. The authors demonstrate that memory B cell evolution can broaden the breadth of neutralizing antibodies. Fu et al. also show that a D94N amino acid substitution in the variable light chain CDR1 improves binding and neutralization activity to the H5 HA subtype.

Minor concerns with this manuscript are as follows:

- Detailed methodology/parameters should be provided for the in silico modeling performed with the Schrodinger software.

This manuscript provides an overall summary regarding broadly neutralizing mAb 3I14. The authors have addressed all of my concerns.

General Reply:

We appreciate the reviewers' thoughtful suggestion and the editor's efforts in helping us improve the manuscript. The manuscript is revised to meet each of the reviewers' suggestion and more details are included. We believe that we have now addressed all issues raised.

REVIEWERS' COMMENTS:

Reviewer #1 (Remarks to the Author):

The authors have extensively revised the manuscript in response to reviewer comments. In particular they add a new analysis of ADCC activity that provides a much better surrogate of cell killing than the reporter assay used in the initial submission.

I have only a few minor remaining comments:

Minor points:

Several reviewer comments are addressed only in the rebuttal letter and not in the text or supplemental material. While I appreciate reading the authors' rationale for choosing the virus strains for in-vivo challenge experiments in the rebuttal, this information should be included in the manuscript itself so that readers of the paper can also understand.

A: We add the details of animal study in Results: Prophylactic efficacy against influenza viruses in mice.

Lines 176-178, "Next, we converted 3I14 into full-length human IgG1 to evaluate protective efficacy in a BALB/c mouse infection model against lethal strains that were available at the time of the study including H5N1, H3N2, H7N7 and H7N9 (Fig. 3)."

Several of reviewer 2's comments are similarly addressed only in rebuttal. For example, it would be helpful for the final manuscript to include information about the FACS gating strategy and exclusion of doublets, and also the comparison of levels of somatic hypermutation in 3I14 and other bnAbs.

A: The FACS gating strategy is shown in Supplementary Fig.10. And the description is added in Method: FACS sorting of H3 binding memory B cells.

Lines 517-519, "Single H3-reactive memory B cells were excluded doublets using SSC and FSC gate (Supplementary Fig. 10), then were sorted into 384-well plate."

The comparison of levels of SHMs in bnAbs is shown in Supplementary Table 3. We also add the description in **Discussion** section.

Lines 347-349, "There are notable similarities and differences among the stem-directed bnAbs encoded by *IGHV1-69* and *IGHV3-30* in their structures, V segment SHMs¹ (Supplementary Table 3) and involvement of light chains in binding."

The authors add a few sentences in the end of the discussion section summarizing various vaccination approaches that have been used to attempt to stimulate bnAbs against influenza. However, this still does not explain why they believe that characterization of 3I14 (or 3I14-expressing memory B cells) per se constitutes a

conceptual advance toward the goal of "universal" influenza vaccination. Similarly, while they have slightly modified the statement in the abstract about an "optimized memory B cell pool," the revised discussion does not explain either the criteria by which memory B cell pools might be considered "optimized" or speculate on how specifically vaccine modalities might do this. The authors should add a few sentences placing 3I14 in context with other bnAbs. They should also either explain how 3I14-like cells represent "optimized" memory B cells or else modify this statement in the abstract.

A: The discussion section is revised to clarify why 3I14-like memory B cells are an "optimized memory B cell pool" and will benefit "universal" influenza vaccination strategies.

Lines 421-430, "3I14-like memory B cells are a general pool that we wish to preserve and expand in that they are not terminally committed to any HA subtype; rather they maintain a plasticity potential that can be evoked by minimal additional SHMs. This is supported by our back-mutation studies that show 3I14 GL antibody has broad activity against H1 and H3 subtypes and has the potential to gain broader activity against H5. This is a distinguishing feature from the GL version of plasma derived FI6 and other bnAbs which greatly loose activity against some subtypes while maintaining strong activity against other subtypes^{2, 3}. We propose an improved vaccination strategy that preferentially targets the HA stem may maintain the longevity of 3I14-like and other anti-stem memory B cells."

Lines 433-435, "Additionally, targeting influenza A vaccines to a naive pool of *IGHV3-30* B cells that provide strong initial binding energy through their BCR H/L CDR3 motifs may prove useful."

Reviewer #2 (Remarks to the Author):

"A broadly neutralizing anti-influenza antibody reveals ongoing capacity of hemagglutinin-specific memory B cells to evolve" presents an overview regarding VH3-30-derived broadly neutralizing antibody 3I14. The authors demonstrate that memory B cell evolution can broaden the breadth of neutralizing antibodies. Fu et al. also show that a D94N amino acid substitution in the variable light chain CDR1 improves binding and neutralization activity to the H5 HA subtype.

Minor concerns with this manuscript are as follows:

- Detailed methodology/parameters should be provided for the *in silico* modeling performed with the Schrodinger software.

A: The details of *in silico* modeling are added in **Method: *in silico* structure modeling**.

Lines 659-665, "3I14 was homology modeled using the antibody-modeling module as implemented in BioLuminate (Schrödinger, Inc). Briefly, the heavy chain and light chain sequence were entered into the program, and the template for framework (FRH/FRL region) was identified by searching against antibody structure database. The PDB file 4GXU was selected as suitable template due to the best composite score (0.94) considering both heavy and light chains. Then the CDR loops-simulated 3I14 model was generated after analyzing of all available antibody loop structures."

This manuscript provides an overall summary regarding broadly neutralizing mAb 3I14. The authors have addressed all of my concerns.

1. Avnir Y, *et al.* Molecular signatures of hemagglutinin stem-directed heterosubtypic human neutralizing antibodies against influenza A viruses. *PLoS pathogens* **10**, e1004103 (2014).
2. Pappas L, *et al.* Rapid development of broadly influenza neutralizing antibodies through redundant mutations. *Nature*, (2014).
3. Corti D, *et al.* A neutralizing antibody selected from plasma cells that binds to group 1 and group 2 influenza A hemagglutinins. *Science* **333**, 850-856 (2011).